

# Estimating global gross primary productivity using chlorophyll fluorescence and a data assimilation system with the BETHY-SCOPE model

Alexander J Norton[1], Peter J Rayner[1], Ernest N Koffi[2], Marko Scholze[3], Jeremy D Silver[1], and Ying-Ping Wang[4]

[1]School of Earth Sciences, University of Melbourne, Melbourne, Australia
[2]European Commission Joint Research Centre, Ispra, Italy
[3]Department of Physical Geography and Ecosystem Science, Lund University, Lund, Sweden
[4]CSIRO Oceans and Atmospheres, Aspendale, Australia

*Correspondence to:* P.J. Rayner (prayner@unimelb.edu.au)

**Abstract.**

This paper presents the assimilation of solar-induced chlorophyll fluorescence (SIF) into a terrestrial biosphere model to estimate the gross uptake of carbon through photosynthesis (GPP). We use the BETHY-SCOPE model to simulate both GPP and SIF using a process-based formulation, going beyond a simple linear scaling between the two. We then use satellite SIF data from the Orbiting Carbon Observatory-2 (OCO-2) for 2015 in the data assimilation system to constrain model biophysical parameters and GPP. The assimilation results in considerable improvement in the fit between model and observed SIF, despite a limited capability to fit regions with large seasonal variability in SIF. The SIF assimilation increases global GPP by 31% to $167 \pm 5 \, \mathrm{PgCyr^{-1}}$ and shows an improvement in the global distribution of productivity relative to independent estimates, but a large difference in magnitude. This change in global GPP is driven by an overall increase in photosynthetic light-use efficiency across almost all biomes and more minor, regionally distinct changes in APAR. This process-based data assimilation opens up new pathways to the effective utilization of satellite SIF data to improve our understanding of the global carbon cycle.

## 1 Introduction

Through photosynthesis terrestrial plants fix atmospheric carbon dioxide ($CO_2$) into organic compounds constituting the largest carbon flux on Earth. This process is the first step in terrestrial carbon sequestration and plays a critical role in offsetting anthropogenic carbon emissions (Campbell et al., 2017; Janssens et al., 2003). However, the gross uptake of $CO_2$ through photosynthesis (GPP; Gross Primary Production) cannot be observed at large spatial scales, which limits our understanding of its spatiotemporal distribution and response to climate (Schimel et al., 2015). Ignorance of GPP limits our ability to predict the terrestrial net $CO_2$ flux under future climate conditions (Friedlingstein et al., 2014; Sitch et al., 2015).

Numerous approaches have been developed to estimate GPP at large scales (see Anav et al., 2015). One approach takes existing observations and merges them with process-based models using model-data fusion ('data assimilation') techniques. Process-based models provide a quantitative description of the current state of knowledge underlying terrestrial biospheric




processes. However, there are large uncertainties in model predictions due to both the model formulation and input parameters. Data assimilation provides an effective way of optimising the input parameters and evaluating the consistency of the model with various observational data, providing insight into the model formulation as well (Rayner, 2010). The Carbon Cycle Data Assimilation System (CCDAS) is one example that has been developed to ingest multiple sources of data (Kaminski et al.,

2013; Koffi et al., 2012; Rayner et al., 2005). More generally, various carbon cycle data assimilation systems have been applied using a range of observational data, including atmospheric $CO_2$, soil moisture, the fraction of absorbed photosynthetically active radiation (FAPAR) and flux tower measurements (see Bacour et al., 2015; Kaminski et al., 2012, 2013; Kato et al., 2013; Macbean et al., 2016; Scholze et al., 2016).

Recent remote sensing measurements of solar-induced chlorophyll fluorescence (SIF) (Frankenberg et al., 2011b; Joiner

et al., 2011) offer a novel insight into the spatiotemporal patterns of GPP (e.g. Duveiller and Cescatti, 2016; Guan et al., 2015; Joiner et al., 2014; Li et al., 2018; Luus et al., 2017). Many studies have shown that SIF correlates strongly with GPP across biome types and generally performs better at tracking GPP than traditional reflectance-based vegetation measurements (e.g. NDVI, EVI) (Joiner et al., 2014; Li et al., 2018; Luus et al., 2017; Walther et al., 2016; Yang et al., 2015).

SIF and GPP are linked at the cellular level through the light reactions of photosynthesis. To initiate the light reactions

of photosynthesis, pigment-protein complexes forming so-called photosystems absorb sunlight energy and convert it into the chemical energy required to power photosynthetic $CO_2$ fixation. This absorbed energy, or excitation energy, has one of three fates. Firstly, excitation energy may be used to drive photosynthetic electron transport, ultimately powering photosynthetic $CO_2$ fixation (Krall and Edwards, 1992), termed photochemical quenching (PQ). Secondly, excitation energy may be dissipated as heat via a range of mechanisms used to protect photosystems against excessive light-induced damage (Demmig-Adams and

Adams III, 2006) collectively termed non-photochemical quenching (NPQ). Finally, excitation energy may be passively emitted from the chlorophyll pigments as chlorophyll fluorescence. During photosynthesis PQ and NPQ are actively regulated by plants to balance energy supply and demand under changing environmental conditions (Porcar-Castell et al., 2014). Chlorophyll fluorescence therefore responds dynamically to changes in the rates of PQ and NPQ, providing a highly useful non-invasive tool to monitor leaf physiological processes (for reviews see Baker, 2008; Govindjee, 1995; Porcar-Castell et al., 2014). Mea-

surements of artificially-induced chlorophyll fluorescence at the leaf level have been used for this purpose for several decades (Govindjee, 1995).

With the use of satellite-based instruments global maps of SIF have also been produced (Frankenberg et al., 2011a; Guanter et al., 2012; Joiner et al., 2011). Parazoo et al. (2014) and MacBean et al. (2018) have used this data to optimise model estimates of GPP. Parazoo et al. (2014) developed a framework to use SIF alongside model estimates to redistribute global

GPP patterns by applying linear scaling between SIF and GPP. They did not, however, optimise model parameters so did not alter model predictive capabilities. MacBean et al. (2018) optimised model parameters of a single model (ORCHIDEE). Following empirical evidence, MacBean et al. (2018) related SIF to GPP using a biome-specific linear scaling. In both cases SIF added useful information and induced large shifts in global GPP. However, SIF was not explicitly modelled and therefore was not compared with the observed SIF to assess performance against the data. Koffi et al. (2015) was the first to combine a

process-based model of SIF with a global terrestrial biosphere model. Koffi et al. (2015) performed global simulations of SIF





and a set of sensitivity tests, demonstrating that the model is capable of utilising the SIF data. Norton et al. (2018) extended this to include a module for prognostic leaf growth. Using this model Norton et al. (2018) quantified how effectively SIF could constrain uncertainties in model parameters and GPP, finding a reduction in uncertainty of global annual GPP of 73%, a result consistent with the model used in MacBean et al. (2018). However, no formal optimisation algorithm was applied.

Using a process-based model of SIF may be important in a data assimilation context. Firstly, while SIF and GPP appear to relate linearly at some spatiotemporal scales, this relationship is ultimately driven by underlying non-linear processes and other variables such as absorbed photosynthetically active radiation (APAR) (Yang et al., 2018). A linear scaling approach is therefore likely to be scale-dependent, whereas a process-based approach can be applied to over a range of spatial and temporal scales. Secondly, assuming a linear scaling between SIF and GPP assumes that SIF relates to biophysical parameters in the same way

as GPP. This is unlikely to be accurate, especially for parameters controlling radiative transfer and fluorescence re-absorption. The linear scaling approach therefore preconditions the assimilation to shift GPP in proportion to SIF, which, as we will show, does not always occur with a process-based approach. Therefore, this study makes an advance on past approaches by simulating SIF explicitly using a process-based model. The aim is to integrate satellite observations of SIF into a data assimilation system to optimize model parameters, assess the performance against the data, and estimate spatiotemporal patterns of GPP globally.

## 2  Methods

Here we outline the steps taken to assimilate SIF into the terrestrial biosphere model BETHY-SCOPE. First, we briefly describe the BETHY-SCOPE model. This model is capable of simulating SIF which provides a means of mapping model variables into the observational space. Second, we outline the quantities that are optimized within the data assimilation system. In this study these quantities are the biophysical parameters of BETHY-SCOPE. Third, we describe the satellite SIF observations used.

Fourth, we outline the optimization algorithm and the method for error propagation. Finally we give a brief description of the specifics of the experimental setup.

### 2.1  BETHY-SCOPE

BETHY-SCOPE is a coupling of the existing models BETHY (Biosphere Energy Transfer Hydrology) (Knorr, 2000) and SCOPE (Soil Canopy Observation, Photosynthesis and Energy fluxes;  van der Tol et al., 2009, 2014) and builds upon the

developments by Koffi et al. (2015) and Norton et al. (2018). The coupling of BETHY and SCOPE enables spatially explicit, plant functional type (PFT) dependent, global simulations of GPP and SIF.

BETHY is a process-based terrestrial biosphere model, which is a key element of the Carbon Cycle Data Assimilation System (CCDAS) (Rayner et al., 2005; Scholze et al., 2007). Full model description details can be found elsewhere (e.g. Rayner et al., 2005; Scholze et al., 2007; Knorr et al., 2010). Briefly, BETHY simulates carbon assimilation and plant and soil

respiration within a full energy and water balance. Although we prescribe leaf area index (LAI) to the model, this version of BETHY has an optional leaf area dynamics module for prognostic LAI as described in Knorr et al. (2010). BETHY represents variability in physiology and ecology of plant classes by 13 PFTs (see Table 1) originally based on classifications by Wilson





and Henderson-Sellers (1985). Each model grid cell may consist of up to three PFTs as defined by their grid cell fractional coverage.

**Table 1.** PFTs defined in BETHY and their abbreviations.

| PFT # | PFT Name | Abbreviation |
|---|---|---|
| 1 | Tropical broadleaved evergreen tree | TrEv |
| 2 | Tropical broadleaved deciduous tree | TrDec |
| 3 | Temperate broadleaved evergreen tree | TmpEv |
| 4 | Temperate broadleaved deciduous tree | TmpDec |
| 5 | Evergreen coniferous tree | EvCn |
| 6 | Deciduous coniferous tree | DecCn |
| 7 | Evergreen shrub | EvShr |
| 8 | Deciduous shrub | DecShr |
| 9 | C3 grass | C3Gr |
| 10 | C4 grass | C4Gr |
| 11 | Tundra vegetation | Tund |
| 12 | Swamp vegetation | Wetl |
| 13 | Crops | Crop |

SCOPE (version 1.53) is a vertically-integrated (1D) radiative transfer and energy balance model with modules for photosynthesis and chlorophyll fluorescence (van der Tol et al., 2009). It utilizes a canopy radiative transfer scheme based on the Scattering by Arbitrarily Inclined Leaves (SAIL) model (Verhoef, 1984) and the leaf radiative transfer model of Fluspect (Miller et al., 2005) which is based upon the optical properties of leaves (Jacquemoud and Baret, 1990). A limitation of this SCOPE version is that it lacks a water balance and only accounts for vertical variation in canopy properties, not horizontal variation. We note that a recent update has included a water balance and water stress in SCOPE, although this was only tested at a semi-arid grassland site (Bayat et al., 2019).

While van der Tol et al. (2009, 2014) provide a more comprehensive description of this model, we provide a brief description of the link between SIF and GPP. It is during the iterative calculation of the thermal radiative transfer and energy balance modules, where photosynthesis and chlorophyll fluorescence quantum efficiency ($\phi_F$) of each canopy element are calculated. This includes the leaf biochemistry module, which simulates the photosynthetic rate as the minimum of two potentially limiting reaction rates (see Collatz et al. (1991) for C3 plants and Collatz et al. (1992) for C4 plants). Inputs to the leaf biochemistry module include APAR, relative humidity, temperature, $CO_2$ concentration, $O_2$ concentration and leaf physiological parameters (e.g. carboxylation capacity). The $\phi_F$ for each canopy element is also calculated within this module. To determine the $\phi_F$, the





fate of absorbed quanta via PQ and NPQ must be determined. First, the photochemical yield ($\phi_P$) is determined from the quantum requirement of the photosynthetic dark reactions, i.e. the electron transport rate ($J_e$), where $J_e$ is calculated as

$$J_e = A_g \frac{C_i + 2\Gamma^*}{C_i - \Gamma^*} \, effcon \tag{1}$$

where $A_g$ is the gross photosynthetic rate (i.e. excluding dark respiration), $C_i$ is the intercellular $CO_2$ partial pressure and $\Gamma^*$
is the $CO_2$ compensation point, and $effcon$ is a variable based on the electron requirements and assumptions on the processes limiting electron transport (in SCOPE, C3 plants $effcon$=1/5; C4 plants $effcon$=1/6). Photochemical yield is determined by the ratio of $J_e$ to the total absorbed flux of electrons ($J_{APAR}$). The $\phi_P$ is calculated, based on the Genty et al. (1989) relationship (see van der Tol et al., 2014), as follows

$$\phi_P = \frac{J_e}{J_{APAR}} = \frac{J_e}{0.5APAR} \tag{2}$$

Where $J_{APAR}$ is the flux of electrons absorbed by photosystem II, assumed to equal to half the APAR. In this study, the APAR driving biochemistry is only that absorbed by chlorophyll, which differs from the original SCOPE model but is consistent with understanding of light-harvesting (Fleming et al., 2012). Eq. 2 imposes the condition that the flux of electrons produced from photochemistry must equal those consumed by the photosynthetic dark reactions (van der Tol et al., 2014). The remaining quanta are distributed between chlorophyll fluorescence and NPQ, where NPQ is further split into constitutive
thermal dissipation (constitutive NPQ) and energy-dependent, regulated thermal dissipation (regulated NPQ).

From the Genty et al. (1989) relationship, the steady-state $\phi_P$ equals the ratio of variable fluorescence to total fluorescence: $\phi_P = (F'_m - F_t)/F'_m$, where $F_t$ is the steady-state fluorescence and $F'_m$ is the maximal fluorescence under a saturating pulse, indicating regulated NPQ. This evolved from decades of research using pulse amplitude modulation fluorescence measurements and theory (Baker, 2008). This relationship can be rearranged to:

$$\phi_F = F'_m(1 - \phi_P) \tag{3}$$

Therefore, to obtain $\phi_F$, a formulation for $F'_m$ is required. Understanding of the mechanisms driving $F'_m$ are not yet sufficient for a process-based model applicable in a canopy-scale steady-state photosynthesis model, however van der Tol et al. (2014) showed that its variability could be captured using an empirical formulation. Here, we use the empirical fit to the drought data (Flexas et al., 2002; van der Tol et al., 2014). It is known that regulated-NPQ is controlled by biochemical feedbacks (Zaks
et al., 2013). It is therefore calculated using an empirically derived equation and a variable that describes the strength of the feedback termed the relative light-saturation of photosynthesis, defined as 1 - $\phi_P/\phi_P^0$ (see van der Tol et al., 2014), where $\phi_P^0$ is the maximum potential photochemical yield with typical values of 0.83 (Björkman and Demmig, 1987). Constitutive NPQ is also calculated, but it is known to be low and relatively constant, although the model does include a high temperature correction (van der Tol et al., 2014). Chlorophyll fluorescence quantum yield can thus be calculated by Eq. 3.





The photosystem I (PSI) and photosystem II (PSII) fluorescence spectra are calculated by the Fluspect module based upon the canopy structure, irradiance, biophysical properties (i.e. leaf composition; pigment concentrations, mesophyll structure, and senescent and dry matter contents), and fluorescence quantum efficiency values for low-light unstressed conditions. Only the PSII fluorescence spectra is adjusted for regulatory feedbacks, as PSI fluorescence is considered to be relatively low and
constant (Porcar-Castell et al., 2014). This is modelled by scaling the PSII spectra with the ratio $\phi_F$ from Eq. 3 (sometimes denoted by $\eta_{II}$) to the low-light unstressed quantum yield (sometimes denoted by $\eta_{II(0)}$). Re-absorption and scattering of fluorescence within the canopy is calculated by a separate routine (van der Tol et al., 2009). This is wavelength-dependent and occurs based on leaf composition and canopy structure. The fluorescence of canopy elements are then numerically integrated over canopy depth and orientation to determine the top-of-canopy SIF, similarly performed for leaf photosynthetic rates to
determine GPP.

Overall, the modelled link between SIF and GPP occurs via the above equations. Therefore, variables (e.g. input parameters, environmental variables) that affect the photosynthetic rate will also affect SIF via $\phi_F$. This includes variables that affect APAR, as APAR is an input to the leaf biochemistry module. However, APAR not only modulates $\phi_F$, but has the additional, perhaps more significant effect of scaling the fluorescence spectra. Furthermore, variables such as leaf composition or canopy
structure can influence the escape probability of fluorescence emission by re-absorption and scattering.

In BETHY-SCOPE, the canopy radiative transfer, energy balance and leaf biochemistry schemes of BETHY have been replaced by the corresponding schemes in SCOPE. The spatial distribution, vegetation (PFT) characteristics and carbon balance are handled by BETHY. SCOPE therefore takes climate forcing (meteorological and radiation data) and spatial information from BETHY and returns GPP, enabling process-based global simulations of GPP and SIF.

## 2.2  BETHY-SCOPE Parameters

In this data assimilation system, the quantities to be optimized are the biophysical parameters that relate to SIF and GPP (see Table A1). Parameters can be either global or spatially differentiated by PFT. PFT-dependent parameters enable differentiation between biophysical traits. Two key parameters for this study, the maximum carboxylation rate at 25°C ($V_{cmax}$) and chlorophyll a/b content ($C_{ab}$) are considered PFT-dependent. The $V_{cmax}$ parameter is used in most process-based terrestrial biosphere
models as it is a parameter of the photosynthesis model of Farquhar et al. (1980). The $C_{ab}$ parameter is a parameter specific to the SCOPE model and an important component of the canopy radiative transfer scheme as it strongly influences both SIF and APAR.

In total there are 41 parameters that are optimized by the data assimilation system. The uncertainty associated with each of these parameters is represented by a Gaussian probability density function (PDF). The mean and standard deviation for
the prior parameters are shown in Table A1. Choice of the prior mean and uncertainty follow those used in previous studies (Kaminski et al., 2012; Knorr et al., 2010; Koffi et al., 2015). For new parameters that are not well characterized (e.g. SCOPE parameters) we assign relatively large prior uncertainties and mean values in line with the default SCOPE parameters and with Koffi et al. (2015) and Norton et al. (2018). An exception is the $C_{ab}$ parameters, which are assigned higher prior values than Norton et al. (2018), more in line with physiological understanding.



Parameters exposed to the data assimilation system are chosen based on previous sensitivity tests such as those performed by Verrelst et al. (2015) and Norton et al. (2018). This includes leaf composition parameters such as $C_{ab}$, leaf dry matter content ($C_{dm}$), and leaf senescent material fraction ($C_s$). Also included are structural parameters such as leaf distribution function parameters ($LIDFa$, $LIDFb$), vegetation height ($hc$) and leaf mesophyll structure, the prior values for these were

obtained from literature values and are assigned to groups of PFTs that we assume have a generally similar structural form (see Table A1). Physiological parameters are also incorporated, including $V_{cmax}$ and Michaelis–Menten constants of Rubisco for $CO_2$ ($K_C$) and $O_2$ ($K_O$). Additionally, the photosynthetic kinetic parameter for the maximum oxygenation rate ($V_{omax}$) is included. Given the uncertainty of $V_{omax}$ and its importance for modelling GPP (von Caemmerer, 2000), this may be an important parameter to consider and is given by its ratio with $V_{cmax}$, $a_{V_o,V_c}$. Given this parameter also affects the relative

specificity of Rubisco ($S_{c/o}$) we calculate $S_{c/o}$ explicitly following von Caemmerer (2000) which differs from the original SCOPE model.

## 2.3 Satellite SIF Observations

We use satellite SIF observations from the NASA Orbiting Carbon Observatory-2 (OCO-2) (Sun et al., 2018). Launched in July 2014, OCO-2 operates in a sun-synchronous orbit with an overpass at approximately 1:30 p.m. local time and a repeat cycle of

16 days. Collecting approximately 24 spectra per second it has relatively high data density within the field of view. OCO-2 has a ground-pixel size of $1.3 \times 2.25$ km$^2$ and a total swath width of 10.6 km. Full spatial mapping of SIF is therefore not possible with OCO-2. However, the high spectral resolution of OCO-2 allows for robust and accurate SIF retrievals (Frankenberg et al., 2014; Sun et al., 2018).

Alternative satellite SIF datasets are also available, including from the GOME-2 and GOSAT instruments (Frankenberg

et al., 2011a; Guanter et al., 2012; Joiner et al., 2011). There are benefits and pitfalls in using these alternative data. For example, GOME-2 and GOSAT provide longer time series going back to 2007 and 2009, respectively. GOME-2 also provides better spatial mapping compared with OCO-2. However, there are known issues of sensor degradation with GOME-2 (Zhang et al., 2018). The advantage of the OCO-2 satellite is that it collects eight times more spectra and has a higher spectral resolution providing more robust and data dense observations (Frankenberg et al., 2014; Sun et al., 2018). We note that a formal compari-

son of these other datasets is outside the scope of this study, but a recent comparison of TROPOMI and OCO-2 showed strong agreement (Köhler et al., 2018).

We use the OCO-2 processed SIF-lite data files. For details on the retrieval algorithm for the SIF data see Frankenberg et al. (2014); Sun et al. (2018). This data is gridded at $2° \times 2°$ spatial resolution, equivalent to the model grid resolution. We exclude soundings collected over water as determined by the corresponding International Geosphere-Biosphere Programme (IGBP)

land classification index (Friedl et al., 2010). We use instantaneous SIF at 757 nm and only soundings taken in nadir mode. Data is also available at 771 nm, however the signal at 757 nm is stronger (Sun et al., 2018) thus we only consider that signal. The annual mean OCO-2 SIF for 2015 is shown below in Fig. 1.

There are potential limitations in using OCO-2 for global mapping due to spatial coverage of the observations and sampling bias of biomes, particularly if it differs from the assumed biome types used in the model. We assessed the similarity between





the sampled IGBP land classification index of the OCO-2 soundings and the BETHY-SCOPE PFTs to evaluate this limitation. Considering the differences in vegetation classifications this is a qualitative test. Qualitatively, the occurrence of IGBP biomes sampling appears to be similar to the BETHY-SCOPE PFTs. Moreover, Frankenberg et al. (2014) showed that despite the limited spatial coverage of OCO-2 it provides a representative sampling of $1° \times 1°$ grid cell averages. We therefore do not

perform any further filtering of the data. Future studies may benefit from evaluating this in more detail and performing the assimilation using observations at a PFT-specific level rather than the grid cell level as is applied here.

### 2.3.1   Observational Uncertainty

The calculation of observational uncertainties is an important aspect of any data assimilation study as it partly determines posterior probabilities. We note two rather extreme cases in calculating the uncertainty in the satellite observations of SIF. The

first is to take the average of the single measurement precision error, considered an overestimate of the uncertainty as it does not account for the sample size. Second is to calculate the standard error, where the average of the single measurement precision error is divided by the square root of the number of observations, as applied in Parazoo et al. (2014). Use of the standard error almost certainly underestimates the uncertainty as it neglects correlated or systematic errors.

Therefore, to determine the measurement error of SIF ($\sigma$) in a given grid cell ($i$), we sum the single measurement precision

error ($\sigma_e$) of each sounding within that grid cell and divide by the total number of soundings ($n_i$). Dividing this by one half scales it closer to the standard error but remains a conservative estimate of the actual error.

$$\sigma = \frac{1}{2} \frac{\sum \sigma_e}{n_i} \tag{4}$$

Calculated uncertainties are shown for January and July 2015 in the supplementary material Figs. S1 and S2. Statistical tests on the results outlined further below will allow us to test whether these observational uncertainties are consistent with other

aspects of this data assimilation process.

### 2.4   Data Assimilation System

To assimilate SIF into BETHY-SCOPE we require a minimization algorithm, cost function, and error propagation method. A variety of techniques are available for the optimization of terrestrial biosphere models and reviews are available (Fox et al., 2009; Kaminski et al., 2013; Macbean et al., 2016; Trudinger et al., 2007).

We utilize a probabilistic framework whereby quantities (e.g. observations, model state variables, model process parameters) are represented by their probability density functions (PDF). These quantities are treated as Gaussian, thus can be described by their mean and standard deviation. For the model parameters the mean is denoted by $x$ and error covariance matrix by $C_x$. We denote the prior parameter vector and covariance matrix by $x_0$ and $C_{x_0}$, respectively, and the posterior parameter vector and covariance matrix by $x_{post}$ and $C_{x_{post}}$, respectively. For the observations the mean is denoted by $d$. The error covariance

matrix in observation space, denoted by $C_d$, combines errors in the observations and in their simulated counterpart i.e. model (Kuppel et al., 2013). Quantification of model error can be performed through an assessment of model-observation residuals

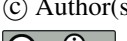



following optimization (e.g. Kuppel et al., 2013). We assess potential model errors in this study, however, we do not explicitly account for this error in the propagation of errors onto GPP hence $C_d$ accounts only for errors in the observations. We point out that the uncertainty is embodied in the error covariance matrices and that diagonal elements represent the variance of the quantities while off-diagonal elements represent error covariances between quantities.

### 2.4.1 Assimilation Procedure

The assimilation procedure finds the posterior PDF for the target variables which, in this case, are the model process parameters. We assume Gaussian PDFs so our posterior PDF is described by its mean and standard deviation. The mean is also the maximum posterior estimate which can be found by minimizing a cost function ($J$). The cost function, shown in Eq. 5, quantifies the difference between the model simulated SIF ($M(x_n)$) and SIF observations ($d$) and the departure of parameter

values ($x_n$) from the prior estimate ($x_0$). These differences are squared and normalized by the uncertainties in the observations $C_d$ model parameters $C_x$, respectively, allowing for more certain quantities to carry more weight. $J$ thus provides a measure of the model-observed mismatch and the deviation from the prior information accounting for uncertainties. We consider the optimization to have converged on an optimal solution when the change in the cost function is less than 1% of the change that occurred during the first iteration.

$$J = \frac{1}{2} \sum \left( \frac{(M(x_n) - d)^2}{C_d} + \frac{(x_n - x_0)^2}{C_x} \right) \tag{5}$$

To find the minimum of $J$ we employ a quasi-Newton method, which is a variational, iterative technique (p. 69 Tarantola, 2005). This algorithm requires a matrix of partial derivatives of the observable with respect to model parameters, called the Jacobian matrix ($H$), calculated using finite differences. $H$ is therefore a representation of the sensitivity of model simulated SIF to each model parameter.

The quasi-Newton algorithm assumes weak non-linearity in the model. This approximation is better than assuming a linear model, but not as useful as having a model adjoint where the entire parameter space can be efficiently examined (Kaminski et al., 2013). With this assumption the model is presumed to be linear about the point where $H$ is calculated. However, to account for non-linearities in the model we recalculate $H$ after each iteration of the algorithm. Given a single 'global' minimum of $J$, this algorithm will converge upon it (Tarantola, 2005). There is still potential that the algorithm will converge upon a

local minimum in $J$.

For each iteration $n$ of the algorithm the parameter vector ($x_n$) is updated using Eq. 6. This adjusts for non-linearity by performing a forward run of the full non-linear model at each iteration ($M(x_n)$). It takes the form:

$$x_{n+1} = x_n - \mu \left( C_{x_0} + H^T C_d^{-1} H \right)^{-1} \left( H^T C_d^{-1} (M(x_n) - d) + C_x^{-1} (x_n - x_0) \right) \tag{6}$$

where $\mu$ is a step-size (set to 0.2) as required in gradient based techniques (Tarantola, 2005). In a case where the parameter

update produces values that are unphysical (e.g. negative $C_{ab}$), they are reset to the nearest physical value for the next iteration.





Alongside $J$ the reduced chi-squared ($\chi_r^2$) statistic is used to assess the match with the observations. Shown in Eq. 7 below, $\chi_r^2$ measures the goodness of fit per observation accounting for observational uncertainties, where $N$ is the number of degrees of freedom which is equal to the total number of observations in our case.

$$\chi_r^2 = \frac{2J}{N} \tag{7}$$

Under the Gaussian assumption this widely applied statistical test assesses the appropriateness of our assumed uncertainties. With a $\chi_r^2$ value of 1 the statistical assumptions that underlie our procedure, including the assumed errors, are consistent with the model-data mismatch (see Tarantola, 1987, p. 212). This means the fit to the data is as good as the assumed distributions say it should be. Informally, this would mean we are neither over-fitting or under-fitting the data.

### 2.4.2 Error Estimation

For linear and weakly non-linear problems Gaussian probability densities propagate forward through to Gaussian distributed quantities (Tarantola, 2005), termed linear error propagation. The posterior parameter errors, $C_{x_{post}}$, are estimated using linear error propagation as shown in Eq. 8 as follows:

$$C_{x_{post}}^{-1} = C_{x_0}^{-1} + H^T C_d^{-1} H \tag{8}$$

where $H$ is calculated at the posterior (i.e. $x_{post}$). Rayner et al. (2005) demonstrated how to propagate these parameter
uncertainties forward through a model onto simulated quantities such as carbon fluxes. Using the Jacobian rule for probabilities, parameter uncertainties in the model parameter covariance matrix ($C_{x_0}$ and $C_{x_{post}}$) can propagate forward onto GPP using Eq. 9: this determines the error covariance of GPP ($C_{GPP}$).

$$C_{GPP} = H_{GPP} C_x H_{GPP}^T \tag{9}$$

Where $H_{GPP}$ is the model Jacobian with respect to GPP. To calculate the prior error covariance of GPP, $H_{GPP}$ is calculated
about the prior parameter vector and $C_x$ equals $C_{x_0}$. To calculate the posterior error covariance of GPP, $H_{GPP}$ is calculated about the posterior parameter vector and $C_x$ equals $C_{x_{post}}$. The difference between these two cases determines the change in GPP error covariance and therefore the uncertainty reduction in GPP.

### 2.5 Experimental Setup

In this study BETHY-SCOPE is run for the year 2015. This constitutes the optimization (or calibration) period. We then
assess the optimized model performance against independent OCO-2 observations outside of the optimization period from September-December 2014.



The model is run on a $2° \times 2°$ grid resolution. Model SIF is calculated at the equivalent wavelength as OCO-2 SIF (757 nm) and overpass time (1:00 - 2:00 p.m. local time). Climate forcing data is provided in the form of daily meteorology (precipitation, minimum and maximum temperatures, and incoming solar radiation) obtained from the WATCH/ERA Interim data set (WFDEI Weedon et al., 2014). These are used to derive average diurnal cycles of climate forcing. Atmospheric $CO_2$ concentration is

set to the 2015 annual average of 397 ppm. LAI is prescribed to the model using the MODIS improved LAI dataset (Yuan et al., 2011). The LAI is averaged at the model $2° \times 2°$ grid resolution and for each grid cell it is split between PFTs using the model PFT grid cell fractional coverage. Photosynthesis and fluorescence are simulated at an hourly time step but forced by the respective monthly mean diurnal cycle such that a single diurnal cycle is simulated for each month.

## 2.6 Global GPP Products for Comparison

To assess the SIF-optimized global GPP we compare the model prior and posterior GPP to other global GPP products. The first dataset for comparison is an upscaled product based on site level measurements termed FLUXCOM GPP (Tramontana et al., 2016). The FLUXCOM GPP product uses various machine learning techniques to empirically upscale flux tower data using remote sensing and meteorological data as the predictor variables. Here, we use the ensemble average of the FLUXCOM GPP product that uses remote sensing data exclusively (Tramontana et al., 2016). The second dataset for comparison is an ensemble

of eleven global dynamic vegetation models forced with equivalent climate fields and atmospheric $CO_2$ concentration that were used to investigate trends in sources and sinks of $CO_2$ (TRENDY; Sitch et al., 2015). It is important to note that global GPP is highly uncertain and that both the FLUXCOM GPP and TRENDY GPP estimates are based on their own model assumptions and/or sparse measured data (Anav et al., 2015). Therefore, these data are used to evaluate whether the SIF assimilation results in global patterns of GPP that align with the current understanding and not for extensive validation purposes.

## 3 Results

There is an abundance of results that may be presented from a global data assimilation study with SIF. First, we present the model fit to the observed SIF for the prior and posterior cases and for the calibration and validation periods. Second, we examine the estimated parameters and their associated uncertainties. Third, we present the spatiotemporal patterns of model GPP alongside other model GPP products. Finally, we present derived quantities from the model including APAR and

photosynthetic light-use efficiency.

### 3.1 Assimilation with SIF

Here we show how the model compares with the observed SIF (shown in Fig. 1) for the prior and posterior cases and for the calibration and validation periods. The goodness of fit between modelled and observed SIF is assessed using multiple metrics. The $\chi_r^2$ fit is a key metric (see Eq. 7). Differences between the model and observations ('residual') and the squared residual

normalized by the observational variance ('mismatch') are also used. The mismatch is a measure of the difference between the model and observations accounting for observational uncertainties, indicating the contribution grid cells make toward the cost

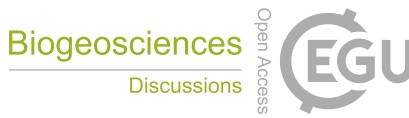



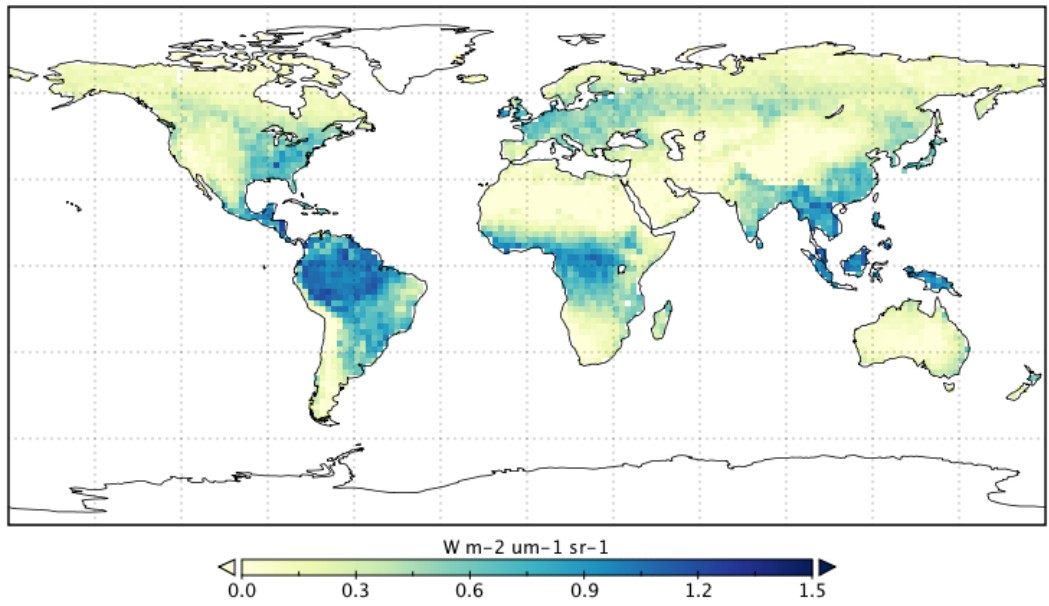

**Figure 1.** Annual mean observed SIF from the OCO-2 satellite for 2015.

function. The model fit during the calibration period is presented in more detail as there is more data. The model fit during the validation period provides a more stringent test of the assimilation performance. We then show the performance of an additional model simulation testing seasonal variation in parameters.

### 3.1.1 Calibration

5   The model prior SIF (SIF$_{\text{prior}}$) over the calibration period yields a global $\chi_r^2$ fit of 2.45. Large residuals in the annual mean, shown in Fig. 2, are evident across the globe, ranging from -0.87 to +1.20 $\mathrm{W m^{-2} \mu m^{-1} sr^{-1}}$. Generally, SIF$_{\text{prior}}$ overestimates observed SIF across regions dominated by tropical forest (e.g. the Amazon, western equatorial Africa and Maritime Continent), boreal forest (parts of North America and Eurasia), and semi-arid regions (e.g. central Australia, central Asia and southern Africa). SIF$_{\text{prior}}$ tends to underestimate observed SIF across the rest of the land, in particular for regions dominated by croplands

10   (e.g. American Midwest, parts of Europe, India, eastern Asia), mixed forests (across Europe and Asia), and grassland and savanna regions (e.g. the African savanna and Brazilian Highlands of South America). Latitudinal averages, shown in Fig. 4, also show these spatial patterns for SIF$_{\text{prior}}$. Overestimation of observed SIF is seen over the central tropics between 15°S and 5°N, a region dominated by tropical evergreen forest (TrEv), whereas there is significant underestimation of observed SIF over the northern hemisphere, particularly during northern summer (see Fig. 5).




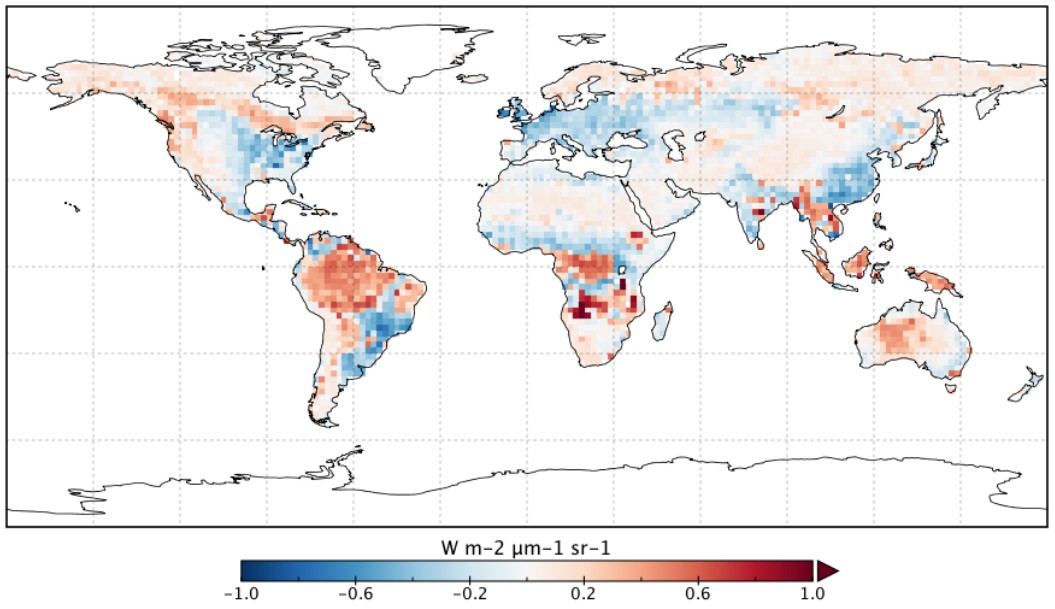

**Figure 2.** Annual mean residual between model SIF$_{\text{prior}}$ and observed SIF for 2015.

Following the assimilation, the model shows a considerably better fit to the calibration data. The global $\chi^2_r$ fit is strongly reduced from 2.45 to 1.01, close to the optimal value of one, demonstrating the ability of the optimized model to fit the observed patterns of SIF and supporting our choices of uncertainty. Annual mean residuals between model posterior SIF (SIF$_{\text{post}}$) and the observations, shown in Fig. 3, range between -0.58 and +0.45 $\mathrm{Wm}^{-2}\mu\mathrm{m}^{-1}\mathrm{sr}^{-1}$, considerably smaller than SIF$_{\text{prior}}$. The spatial

patterns of posterior residuals (Fig. 3) show that the regional patterns of residuals broadly exhibit the same sign as the prior case albeit with a significantly reduced magnitude. For the latitudinal averages, SIF$_{\text{post}}$ is remarkably close to the observed SIF for the annual average (Fig. 4). However, discrepancies are evident for the northern summer averages for both SIF$_{\text{prior}}$ and SIF$_{\text{post}}$ (Fig. 5), where observed SIF is underestimated in the northern hemisphere and overestimated in the southern hemisphere.

Latitudinal sums of the mismatch between the model and the observations (bar charts in Figs. 4 and 5) also show a significant

reduction (i.e. improvement in fit) following the assimilation. Comparison of the gray and green bars, indicating the respective prior and posterior mismatch, show that this improvement in fit occurs over all latitudes at the annual and northern summer time scales. The total annual mismatch between SIF$_{\text{post}}$ and the observations is about 40-80% smaller across the latitudes between 40°S to 60°N relative to SIF$_{\text{prior}}$.

Despite the strong improvement in fit, SIF$_{\text{post}}$ tends to underestimate large observed SIF values >1.0 $\mathrm{Wm}^{-2}\mu\mathrm{m}^{-1}\mathrm{sr}^{-1}$,

shown in the supplementary material Fig. S4. A linear regression line between the observed SIF and SIF$_{\text{post}}$ has a slope of 0.67. Furthermore, while the observed SIF for any given month and grid cell can reach up to 2.29 $\mathrm{Wm}^{-2}\mu\mathrm{m}^{-1}\mathrm{sr}^{-1}$, SIF$_{\text{prior}}$





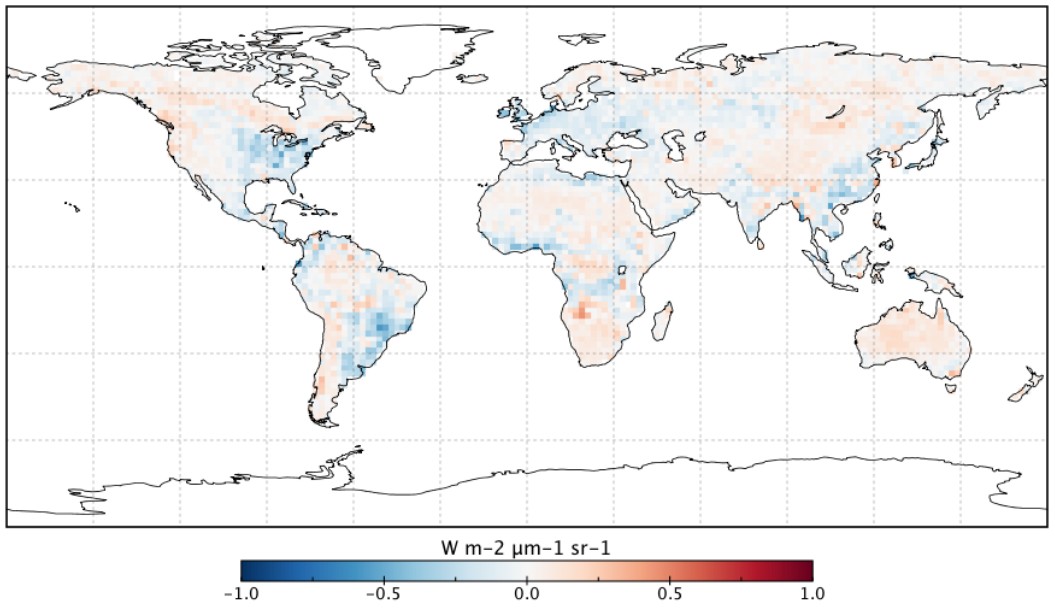

**Figure 3.** Annual mean residual between model SIF$_{post}$ and observed SIF for 2015.

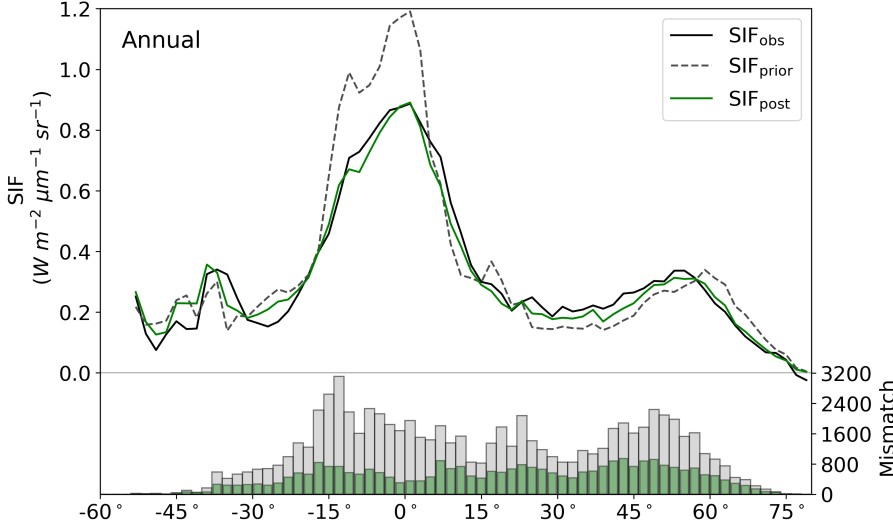

**Figure 4.** Latitudinal averaged SIF and mismatch with observations. OCO-2 observations (black line), SIF$_{prior}$ (gray dashed line, gray bars), and SIF$_{post}$ (green line, green bars) for the annual average. Data is only shown for spatiotemporal points where OCO-2 observations are present.





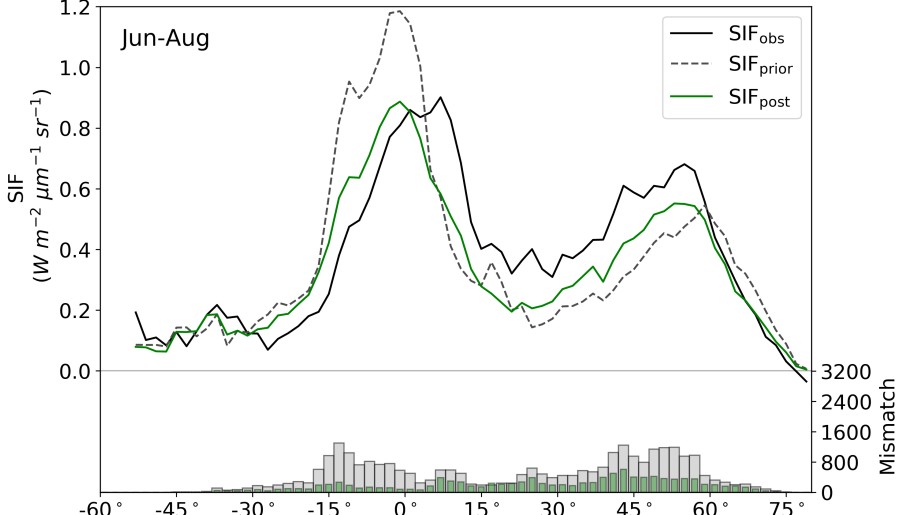

**Figure 5.** Latitudinal averaged SIF and mismatch with observations. OCO-2 observations (black line), SIF$_{prior}$ (gray dashed line, gray bars), and SIF$_{post}$ (green line, green bars) for northern summer (June-August). Data is only shown for spatiotemporal points where OCO-2 observations are present.

does not exceed 1.98 Wm$^{-2}$μm$^{-1}$sr$^{-1}$ and SIF$_{post}$ does not exceed 1.43 Wm$^{-2}$μm$^{-1}$sr$^{-1}$. The SIF$_{prior}$ does not show this systematic underestimation however, but it has a poorer global fit (supplementary material Fig. S3). We note that these large observed SIF values occur mostly over the tropics and the northern mid-latitudes during the peak growing season.

From Fig. 3 it appears that SIF$_{post}$ overestimates observed SIF over arid regions (e.g. the Sahara, Atacama, and Namib deserts,
5   central Australia and central Asia). This is largely because of observed SIF values that are slightly negative, potentially due to measurement noise or issues from the correction of constant error artifacts in the SIF retrieval (Sun et al., 2018). Negative SIF values are still considered in the assimilation system. However, they contribute little to the overall mismatch given the uncertainty in the SIF observations (see supplement Figs. S7 and S8).

### 3.1.2   Validation

10   To validate the optimized model we assess the model fit to independent OCO-2 SIF data from September-December 2014, outside of the calibration period. The global $\chi_r^2$ for SIF$_{prior}$ is 2.57, while SIF$_{post}$ is 1.06. This indicates a strong improvement in fit following the SIF assimilation. Comparison of SIF$_{post}$ with the validation data shows that large SIF values (>1.0 Wm$^{-2}$μm$^{-1}$sr$^{-1}$) are systematically underestimated, similar to the fit to calibration data. For this validation data, these large SIF values typically occur over tropical forest, grassland and cropland regions.





### 3.1.3 A Case with Seasonally Varying Parameters

Most terrestrial biosphere models assume process parameters are constant through time despite evidence showing that some of these biophysical variables (e.g. $C_{ab}$, $V_{cmax}$) vary in response to resource availability (e.g. Demarez, 1999; Wang et al., 2007; Wilson et al., 2000; Xu and Baldocchi, 2003; Zhang et al., 2014). At present the BETHY-SCOPE model does not include any

mechanism for varying these with time, other than a temperature correction for $V_{cmax}$. Given this, we expect that assuming these parameters are temporally constant will contribute to the disparity between the modelled and observed SIF, particularly for more seasonal vegetation.

Thus, an additional comparison is made where we apply a simple seasonal cycle to $C_{ab}$ and $V_{cmax}$ parameters for the posterior model. We set the annual mean to be the posterior $C_{ab}$ and $V_{cmax}$ values and apply a seasonal cycle by using a sine

function that has a period of one year, a maximum on the summer solstice (i.e. December 22nd in southern hemisphere and June 22nd in northern hemisphere) and an assigned amplitude. For highly seasonal PFTs including deciduous trees and shrubs, C3 and C4 grasses, and crops, the amplitude is set to 50% of the mean, while for all other PFTs the amplitude is set to 10%. While this seasonal cycle is arguably oversimplified, this still provides us with a simple sensitivity test to investigate whether introducing a more formal seasonal variation in $C_{ab}$ and $V_{cmax}$ would improve the fit with the observed SIF.

Implementation of seasonally varying $C_{ab}$ and $V_{cmax}$ results in a moderate improvement in fit with the observed SIF. The posterior $\chi_r^2$ fit improves from 1.01 to 0.89 given the seasonally variable parameters (SIF$_{post,seas}$). Both the coefficient of determination ($R^2$) and slope of a linear regression line improve with SIF$_{post,seas}$; with $R^2$ increasing from 0.74 to 0.77 and the slope increasing from 0.67 to 0.71. This indicates that the systematic underestimation of large observed SIF values may be improved. Furthermore, the $\chi_r^2$ fit to the observed SIF data >1.0 $\mathrm{Wm}^{-2}\mu\mathrm{m}^{-1}\mathrm{sr}^{-1}$ shows a strong improvement given seasonally varying

parameters, going from 3.97 for SIF$_{post}$ to 2.99 for SIF$_{post,seas}$. The $\chi_r^2$ fit to low SIF values (<0.25 $\mathrm{Wm}^{-2}\mu\mathrm{m}^{-1}\mathrm{sr}^{-1}$) also improves from 0.56 for SIF$_{post}$ to 0.50 for SIF$_{post,seas}$. Notably, when the fit is assessed per PFT (i.e. grid cells with the same spatially dominant PFT are considered together) the $\chi_r^2$ fit improves for all of these 'biomes'.

### 3.1.4 Fit to the Seasonal Cycle

We can also assess the seasonal cycle of SIF to determine how well the model simulates the amplitude of observed SIF. First,

we assess how well the model replicates the seasonal amplitude of observed SIF across all spatial points. Second, we assess the seasonal patterns of SIF for a selection of case study regions in more detail. We avoid assessing the seasonal cycle of SIF aggregated at global or hemispheric scales as regional patterns of residuals can differ in sign and magnitude (e.g. see Fig. 3). The seasonal amplitude of observed SIF is calculated as the difference between the maximum and minimum SIF across the year for each grid point. To increase confidence that the observations really capture the seasonal cycle, we only assess grid

points with at least eight months of observed SIF data. In doing so, most regions north of 60°N are excluded due to limited SIF observations. We do not assess the timing of the seasonal cycle (e.g. start and end of the growing season) as this is largely driven by LAI which is prescribed and therefore fixed in this study.





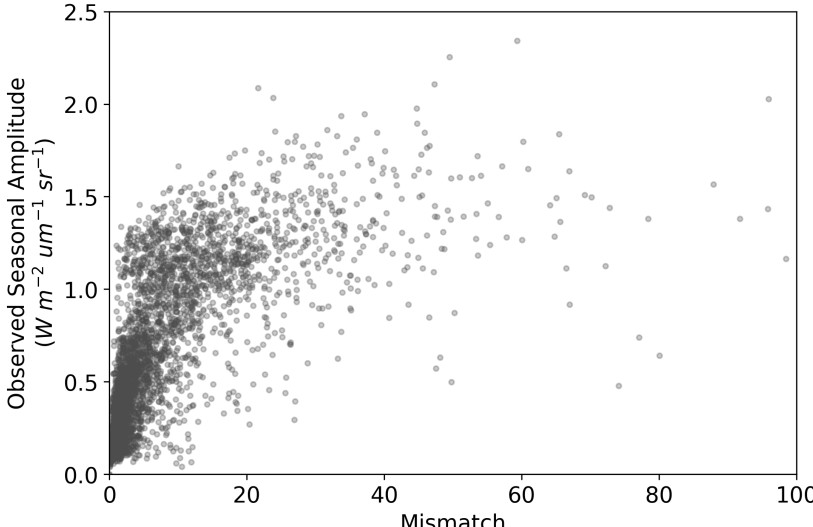

**Figure 6.** Seasonal amplitude of observed SIF versus the model-observed mismatch for the SIF optimized (SIF$_{post}$) model. Mismatch is defined as the squared residual normalized by the observational uncertainty (expressed as variances). A positive curvi-linear relationship exists such that spatiotemporal points with a large model-observed mismatch generally also exhibit a large observed seasonal amplitude.

The comparison of the seasonal amplitude of observed SIF against SIF$_{prior}$, SIF$_{post}$ and SIF$_{post,seas}$ is shown in the supplementary material Fig. S9. The model underestimates the observed seasonal amplitude in all cases. With a perfect match to the observed seasonal amplitude the model would follow the 1:1 line and the slope of a linear regression line would equal 1. However, we find that the slope is 0.21 for SIF$_{prior}$, 0.35 for SIF$_{post}$, and 0.43 for SIF$_{post,seas}$. Spatial points with the largest seasonal variations in observed SIF also exhibit the largest model-observed mismatch (Fig. 6).

For more detailed assessment of seasonal patterns we investigate three case study regions: (i) the tropical forest of mainland south-east Asia; (ii) croplands in North America, and; (iii) the north African savanna (see supplementary material Figs. S10-S15 for details). These regions are selected as they represent quite different biome types, exhibit varied SIF patterns, and have relatively large posterior model-observed mismatch.

The tropical evergreen forest of mainland south-east Asia exhibits a clear seasonal cycle in observed SIF. The monthly mean observed SIF averaged over this region varies from a minimum of ~0.6 $\mathrm{Wm}^{-2}\mathrm{\mu m}^{-1}\mathrm{sr}^{-1}$ in March to a maximum of ~1.4 $\mathrm{Wm}^{-2}\mathrm{\mu m}^{-1}\mathrm{sr}^{-1}$ in August (see Fig. S10). Both SIF$_{prior}$ and SIF$_{post}$ exhibit seasonal cycles that differ strongly from the observations, with little change in the shape of the seasonal cycle from the prior to posterior simulations. Model SIF shows a minimum in July and two maximums in February and November, following the seasonal evolution of LAI (see Fig. S10) which we reiterate is prescribed. This results in strong negative temporal correlations between observed SIF and model SIF as over this region.

Croplands in North America are heavily managed landscapes with highly productive vegetation as indicated by the large observed SIF values during the growing season. Even with the monthly averages used here, observed SIF can exceed 2.0





$\mathrm{Wm}^{-2}\mathrm{\mu m}^{-1}\mathrm{sr}^{-1}$. As presented earlier, the model SIF cannot match the seasonal amplitude of observed SIF and subsequently underestimates the maximum monthly SIF averaged over this region by almost 40% (0.45 $\mathrm{Wm}^{-2}\mathrm{\mu m}^{-1}\mathrm{sr}^{-1}$) (see Fig. S12). The fit is improved in $\mathrm{SIF}_{\mathrm{post,seas}}$ ($\chi_r^2$ ($\mathrm{SIF}_{\mathrm{post}}$) = 4.62; $\chi_r^2$ ($\mathrm{SIF}_{\mathrm{post,seas}}$) = 3.10). In both cases the timing of seasonal maximum and senescence is simulated quite well, while the onset of the growing season is predicted to be too early.

5    The north African savanna exhibits a strong seasonal cycle in observed SIF. This region is dominated by grasslands and open forest, with a seasonality closely following the seasonal variation in precipitation. Averaged over the region, observed SIF varies from 0.07 $\mathrm{Wm}^{-2}\mathrm{\mu m}^{-1}\mathrm{sr}^{-1}$ in January to 0.77 $\mathrm{Wm}^{-2}\mathrm{\mu m}^{-1}\mathrm{sr}^{-1}$ in September (see Fig. S14). However, $\mathrm{SIF}_{\mathrm{post}}$ exhibits a much smaller seasonality with variation from 0.24 to 0.48 across the year, only 34% of the observed seasonal amplitude. Temporal correlations are quite strong however, as model SIF also reaches its peak in September.

## 3.2    Estimated Parameters

The prior and posterior parameter mean values and associated uncertainties are shown in Table A1. In this data assimilation system the number of observations far outweighs the number of unknowns. This means that there is a substantial amount of observational information available to constrain parameter values, thus they can shift from their prior values considerably even if given a relatively tight prior uncertainty. We can be more confident in parameters that see large reductions in uncertainty. Conversely, parameters with little reduction in uncertainty following optimization should be accepted cautiously.

Posterior $V_{cmax}$ estimates range from 16 to 130 $\mathrm{\mu mol m}^{-2}\mathrm{s}^{-1}$. The lowest posterior rates occur for the TmpEv, C4Gr, Tund, and Wetl PFTs, all equal or below 30 $\mathrm{\mu mol m}^{-2}\mathrm{s}^{-1}$. The highest posterior rates occur for the Crop and C3Gr PFTs, both exceeding 100 $\mathrm{\mu mol m}^{-2}\mathrm{s}^{-1}$. Nine out of thirteen PFTs see an increase in $V_{cmax}$. Increases greater than two standard deviations occur for the PFTs TmpDec, EvCn, C3Gr, C4Gr, and Tund, many of which dominate temperate regions. Latitudinal averages of $V_{cmax}$ (supplementary material Fig. S19) show a strong increase in temperate zone $V_{cmax}$ following the SIF assimilaiton, shifting it higher than $V_{cmax}$ in the tropics. Zonally, the lowest $V_{cmax}$ occurs in the cold-climate high latitudes. Most of these parameters see moderately strong uncertainty reductions (>30%) indicating strong constraint from SIF.

Posterior $C_{ab}$ estimates range from 8 to 38 $\mathrm{\mu g cm}^{-2}$. Nine out of thirteen PFTs see a decrease in $C_{ab}$. The highest posterior $C_{ab}$ values occur for the TrEv, TmpEv, TmpDec, DecCn, and Crop PFTs, all >25 $\mathrm{\mu g cm}^{-2}$ while the lowest values occur for the EvShr, Tund, and Wetl PFTs, all <15 $\mathrm{\mu g cm}^{-2}$. Uncertainty reduction ranges from weak to moderately strong for the $C_{ab}$ parameters, with a maximum constraint of 34%. Latitudinal averages of posterior $C_{ab}$ show a relatively low variance across different zones (supplementary material Fig. S20), with posterior values being similar between the tropics and temperate mid-latitudes, albeit with a distinct dip in drier sub-tropics and high latitudes. The leaf composition parameter for dry matter content, $C_{dm}$, increases by about 70% and an uncertainty reduction of about 20%. The leaf composition parameter for senescent material remains almost unchanged and shows only a very weak uncertainty reduction of about 1%.

Parameters that control canopy structure and the leaf angle distribution see large deviations from their prior values. Some leaf angle distribution parameters, LIDFa and LIDFb, shift considerably. SIF is particularly sensitive to the LIDFa and LIDFb parameters, although this depends on which group of PFTs they pertain to, so generate uncertainty reductions of up to 90%. Vegetation height for both trees and shrubs remain relatively unchanged, while vegetation height for grasses and crops see a





decrease of over four standard deviations. However, the uncertainty reduction of the vegetation height parameters is very weak (<1%). Despite these changes GPP is relatively insensitive to the canopy structure parameters.

## 3.3 Estimated GPP

The spatial patterns of posterior GPP and the changes following the SIF assimilation are shown in Figs. 7-11. Following the
assimilation of SIF global GPP increases by about 39 $\mathrm{PgCyr^{-1}}$ from 127.6 $\mathrm{PgCyr^{-1}}$ to 166.7 $\mathrm{PgCyr^{-1}}$. This change shifts BETHY-SCOPE further from the TRENDY mean (142.4 $\mathrm{PgCyr^{-1}}$) and FLUXCOM GPP (103.3 $\mathrm{PgCyr^{-1}}$) estimates. The parametric uncertainty in global GPP is reduced by 38% from ±7.4 $\mathrm{PgCyr^{-1}}$ to ±4.6 $\mathrm{PgCyr^{-1}}$ by the SIF assimilation.

Spatially, increases in annual GPP are seen across much of the land surface as shown in Fig. 8 (also see supplementary material Fig. S16 for the percentage change in GPP). We note here that these changes in model GPP can differ in sign and
magnitude from the changes in model SIF as a result of the non-linear, process-based approach (see supplementary material Fig. S17 and S18). For example, the wet tropical forests (TrEv) and cold-climate conifer forests (EvCn) see an increase in GPP but decline in SIF. Declines in GPP are only seen in dry tropical forests in parts of South America, Africa and mainland Asia (TrDec PFT; also shown in 9). Globally, TrDec-dominated forests see a decline in total GPP of 23%. Two other biomes show a decline in global GPP (see Fig. 9), TmpEv and DecShr. All other biomes see an increase in their global GPP. The global GPP
of TrEv forest biomes, including the Amazon, equatorial Africa and Maritime continent, increase by about 20%. In relative terms, the largest increases in global GPP occur for TmpDec, EvCn, C3Gr, and C4Gr, all >60%.

Spatially, increases in annual GPP are seen across much of land surface as shown in Fig. 8 (also see supplementary material Fig. S16 for the percentage change in GPP) with the exception of dry tropical forests in parts of South America, Africa and mainland Asia. These decreases are due to a decrease in the GPP of tropical deciduous forest biomes (TrDec) as shown in Fig.
9. Globally, TrDec-dominated forests see a decline in GPP by 35%. Two other biomes show a decline in global GPP (see Fig. 9), TmpEv and DecShr. All other biomes see an increase in their global GPP. The global GPP of TrEv forest biomes, including the Amazon, equatorial Africa and Maritime continent, increase by about 20%. In relative terms, the largest increases in global GPP occur for TmpDec, EvCn, C3Gr, and C4Gr, all >60%.

Averaged over latitudinal bands (Figs. 10 and 11) we can see the distinctive peak in GPP across the tropics at the annual
time scale and secondary peak in GPP across the northern mid-latitudes during the northern summer. With the assimilation of SIF the GPP across all latitudes increases. Some regions and seasons see larger changes. The central tropics (15°S-5°N; dominated by the PFT TrEv) increases substantially. For this region, the prior and posterior estimates are near the high end of other estimates, with the posterior GPP exceeding the 90th percentile of TRENDY models (blue shading). FLUXCOM GPP is very low in the central tropics, but we note that this product is not expected to be representative of the tropics given
the sparsity of the flux tower network there (Tramontana et al., 2016). The northern extratropics (30°-60°N) show a general increase in BETHY-SCOPE GPP, with the SIF assimilation shifting it to the higher end of other estimates. This is particularly strong during northern summer (Fig. 11). While the prior GPP in this region is within the TRENDY model range and close to the FLUXCOM GPP, the SIF assimilation results in a posterior that exceeds the 90th percentile of the TRENDY model range. North of 65°N the prior closely matches FLUXCOM GPP, but the posterior sees an increase which brings it more in





line with the TRENDY model average. There is also a distinct difference between the FLUXCOM GPP product and all models north of 75°N, with FLUXCOM GPP being higher. BETHY-SCOPE posterior GPP over the southern latitudes south of 15°S is generally within the TRENDY model range. In this region, the prior GPP is near the bottom of the TRENDY model range, although this shows closer similarity to the FLUXCOM GPP.

A useful metric for patterns of global productivity is the ratio of GPP between different regions. These ratios are summarised in Table B1. The ratio of the tropics (30°S-30°N) to the extratropics (south of 30°S and north of 30°N) declines following the SIF assimilation, due to an large increase in extratropical GPP and relatively smaller increase in tropical GPP. This shifts the ratio of tropical:extratropical GPP from a prior of 2.47 to a posterior of 1.94, which is substantially closer to patterns of the FLUXCOM (1.90) and TRENDY mean (1.93). Similarly, the ratios of the tropics to the boreal region (north of 55°N), tropics

to the temperate region (south of 30°S and north of 30°-55°N), and temperate to boreal region converge toward the FLUXCOM values (see Table B1).

We also note an improvement in the correlation between the BETHY-SCOPE estimate and the FLUXCOM GPP over North America (see Appendix Figs. B5 and B6) and Europe (data not shown), two regions where FLUXCOM GPP has considerably more training data and thus where we expect it to better represent actual GPP. Over North America the correlation improves

from a prior $R^2$=0.80 to a posterior of $R^2$=0.89, while over Europe the correlation improves from a prior $R^2$=0.76 to a posterior of $R^2$=0.86. Despite this improvement in match between the patterns, the posterior slope is 1.6 for North America and 1.8 for Europe indicating that the magnitude of posterior monthly GPP is larger than that of FLUXCOM GPP.

Changes in GPP due to changes in parameter values, can be broken down into changes in intercepted radiation (APAR) and canopy photosynthetic light-use efficiency ($\text{LUE}_{\text{GPP}}$). The $\text{LUE}_{\text{GPP}}$ is calculated as the annual average of the ratio between

monthly GPP to monthly APAR. Overall, the majority of biomes see an increase in $\text{LUE}_{\text{GPP}}$ following the SIF assimilation, shown in Fig. 12. Just three biomes, TrDec, TmpEv, and DecShr, see a decline in $\text{LUE}_{\text{GPP}}$. If we map the percentage change in $\text{LUE}_{\text{GPP}}$ (see Appendix Fig. B2), the regional changes in $\text{LUE}_{\text{GPP}}$ mirror those in the annual mean GPP (Fig. 8). This can be related back to the general increase in $V_{cmax}$ for most PFTs (Table A1) and latitudes (supplementary material Fig. S19). Changes in APAR are smaller in relative terms and show distinct regional differences (Appendix Fig. B4). With the

exception of low-productivity arid regions, the largest percentage change in APAR occurs for the high latitude tundra biome with approximately a 20% increase in APAR, due primarily to an increase in $C_{ab}$. Wet tropical forests see a decline in APAR of about 5%, while drier tropical biomes (e.g. Brazilian Highlands, north African savanna) see an increase of <5%. Other regions show only minor shifts in APAR.

## 4   Discussion

The use of satellite-derived SIF in a data assimilation system has substantially improved the performance of the BETHY-SCOPE model against calibration and validation SIF observations. The posterior model fit is similarly good between the validation period ($\chi_r^2 = 1.06$) and calibration period ($\chi_r^2 = 1.01$), indicating that the model performs similarly well outside of the assimilation period. We show that with the inclusion of seasonal variations in biophysical quantities (e.g. $C_{ab}$ and $V_{cmax}$)





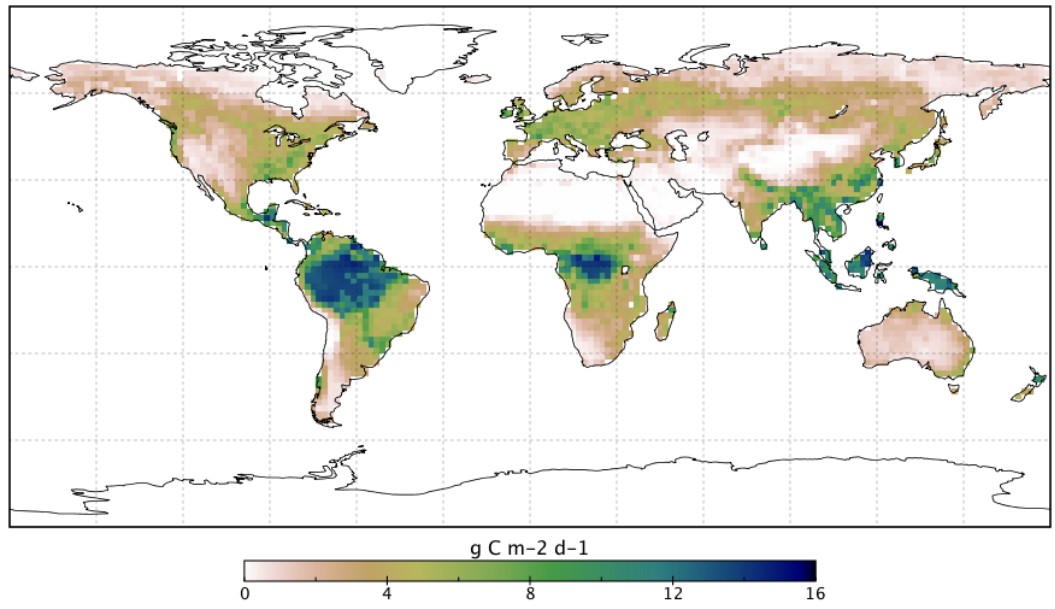

**Figure 7.** Spatial patterns of BETHY-SCOPE posterior annual mean GPP (GPP$_{post}$) for 2015.

the fit to SIF can be improved further as well as provide a better representation of ecosystem function. We highlight that the improvement in fit following the assimilation occurs given equivalent LAI fields. Overall, assessing the optimized model in this way is a key validation test and highlights the improvement following the assimilation. While this is the most stringent validation we can carry out with the available data (considering the available OCO-2 and model climate forcing data), future

work should consider longer periods to sample more varied climatic conditions. Assessment against other satellite SIF products (e.g. GOME-2, GOSAT, TROPOMI) is also feasible provided that careful consideration is taken of the instrumental differences.

The SIF-optimized model produces a global GPP of 166.7 $\mathrm{PgCyr}^{-1}$ for 2015. This is an increase of 31% relative to the prior and is largely due to an increase of GPP in both tropical and extratropical regions. Other approaches to quantify GPP globally have produced a large range of estimates over different periods including 119 $\mathrm{PgCyr}^{-1}$ (Jung et al., 2011), 146 $\mathrm{PgCyr}^{-1}$

(Koffi et al., 2012), 157 $\mathrm{PgCyr}^{-1}$ (Peylin et al., 2016), and 175 $\mathrm{PgCyr}^{-1}$ (Welp et al., 2011). Validating the posterior GPP estimate at these large scales is highly challenging and will require further analysis. Nevertheless, the substantial improvement in fit with SIF data during the calibration and validation periods provides some confidence in the overall spatial patterns (Figs. 4, 5, B1-B4). Indeed, the correlation of BETHY-SCOPE GPP with the FLUXCOM GPP over North America and Europe, regions with many calibration sites, improves with the assimilation of SIF observations despite showing a higher magnitude

suggesting an improvement in spatial patterns of GPP following the assimilation of SIF. There is emerging evidence that the productivity of the northern mid-latitudes is higher than previously thought. This includes evidence from near-infrared





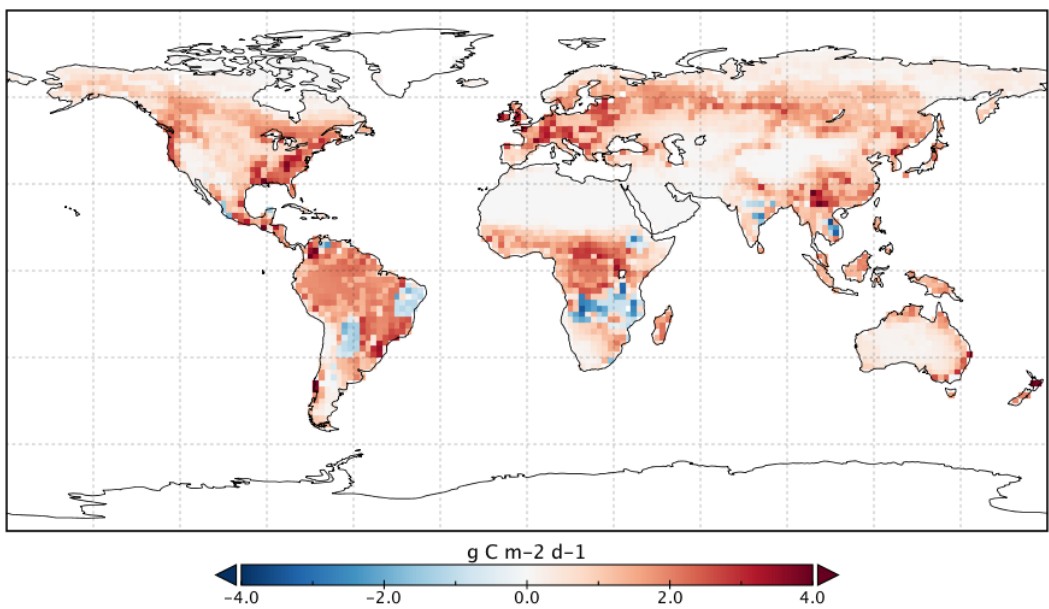

**Figure 8.** Change in annual mean GPP rate for 2015 following optimization with SIF relative to GPP$_{prior}$.

reflectance of vegetation or $NIR_v$ (Badgley et al., 2018) and SIF-based estimates showing that croplands, such as across the American Midwest, have higher GPP than most models predict (Guanter et al., 2014).

The SIF assimilation also alters the distribution of global GPP, with the ratio of the tropics to extratropical regions and the temperate to boreal zone coming into better agreement with FLUXCOM GPP (Table B1). Previous studies that used SIF to constrain model GPP using linear scaling factors between the two have found similar results (Parazoo et al., 2014; MacBean et al., 2018). In both of these studies an increase in tropical GPP was found, which is in agreement with our finding of high tropical GPP. Our tropical GPP estimate exceeds both the FLUXCOM GPP and TRENDY model average. Given the sparsity of the flux tower network, we should expect that the upscaling inherent in the FLUXCOM product is less reliable in the tropics than the mid-latitudes.

The uncertainty reduction from the SIF assimilation is weak to moderate for leaf composition parameters, moderate for canopy structure parameters, and moderate for leaf physiological parameters. The SIF-constraint on these parameter uncertainties results in a moderate overall reduction of parametric uncertainty in global annual GPP of 38%. This differs from previous work that found an uncertainty reduction in global GPP of 73% using a different version of the same model (Norton et al., 2018) and 83% using a different model (MacBean et al., 2018) which could be due to a number of reasons. Firstly, compared with Norton et al. (2018) we use prescribed LAI rather than a prognostic LAI module. Parameters that control LAI were found to be effective at propagating information from SIF to GPP (Norton et al., 2018). The choice to use prescribed rather than prognostic





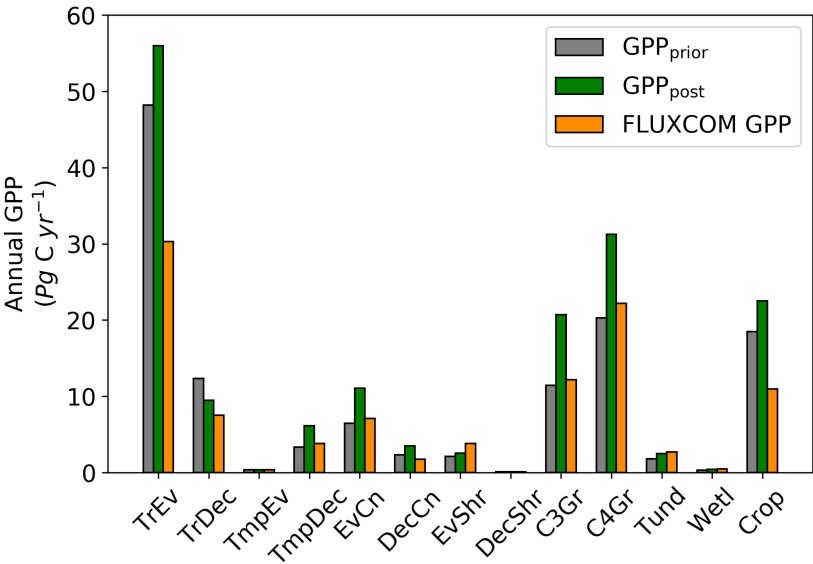

**Figure 9.** Annual GPP for 2015 per biome. Biomes are defined by aggregating model grid cells that have the same spatially dominant PFT as shown in the Appendix Fig. A1.

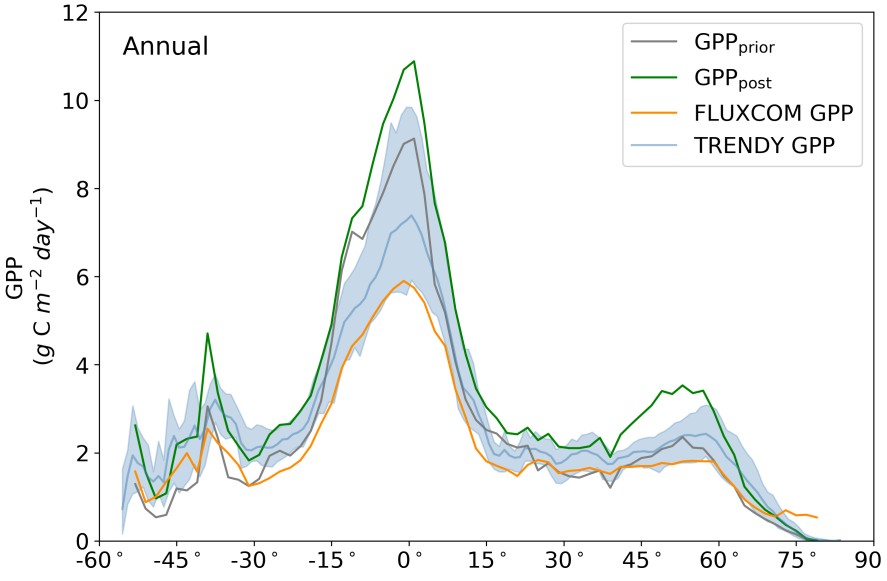

**Figure 10.** Annual latitudinal averages of GPP$_{prior}$ (gray line), GPP$_{post}$ (green line), FLUXCOM GPP (orange line), TRENDY model average (light blue line) and TRENDY model spread given by the 10th and 90th percentiles (light blue shading).





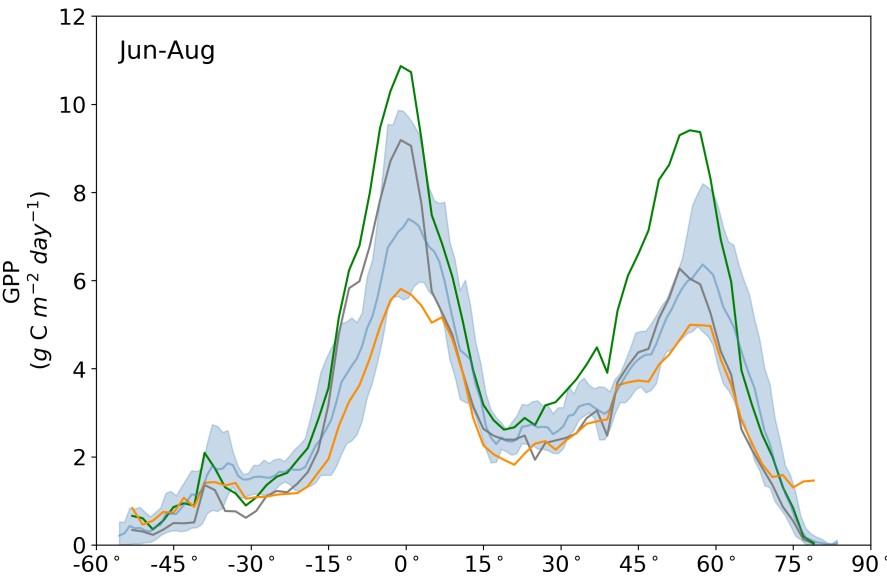

**Figure 11.** Northern summer (June-August) latitudinal averages of GPP$_{prior}$ (gray line), GPP$_{post}$ (green line), FLUXCOM GPP (orange line), TRENDY model average (light blue line) and TRENDY model spread given by the 10th and 90th percentiles (light blue shading).

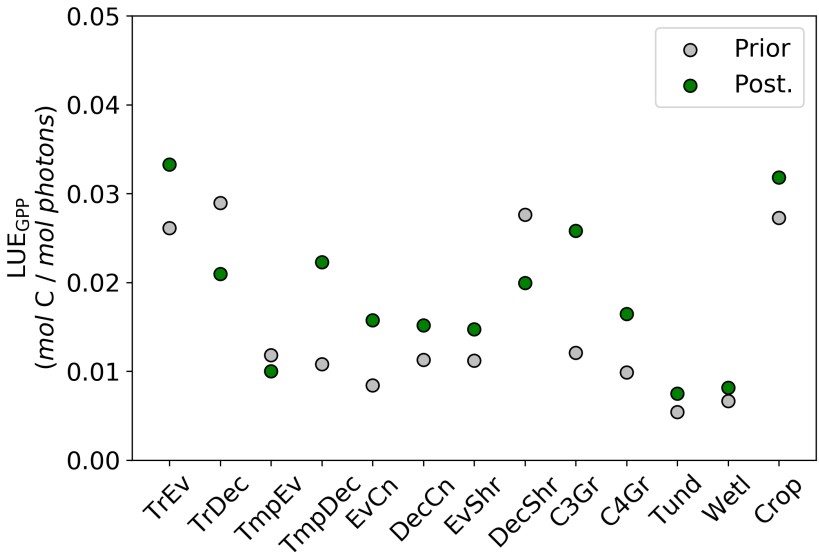

**Figure 12.** Per PFT LUE$_{GPP}$ for the prior and posterior simulations. Values were determined by monthly average GPP divided by monthly averaged APAR, then averaged to annual scales thus averaging over seasonal variations. Note that the theoretical maximum is 0.08 mol C mol photons$^{-1}$ (Waring et al., 2016).



LAI was made due to clear issues with the model simulated LAI, an issue outside the scope of this study. Secondly, there is a much larger constraint of $V_{cmax}$ and smaller constraint of $C_{ab}$ found here compared to Norton et al. (2018). This is likely due to the larger prior $C_{ab}$ values and lower prior uncertainties applied here, chosen because it shifts PFTs out of the range where $C_{ab}$ would strongly limit photosynthetic rate (Björkman, 1981); a quality not expected to be prevailing under natural

conditions (Hirose and Werger, 1987). GPP is strongly sensitive to $C_{ab}$ only when light strongly limits photosynthetic rate, which can occur when $C_{ab}$ content is very low e.g. <15 µg cm$^{-2}$ (Björkman, 1981). Under these conditions any SIF-constraint of $C_{ab}$ will propagate onto GPP quite effectively. However, in this study, more physically defensible $C_{ab}$ values result in other parameters, including $V_{cmax}$, to become relatively more important in simulating SIF, hence, $V_{cmax}$ parameters here exhibit a stronger constraint from SIF compared to Norton et al. (2018). This also suggests that SIF may provide good constraint on GPP

under both light-limited and light-saturating conditions. Additionally, in MacBean et al. (2018) a much stronger constraint of $V_{cmax}$ was found as there was no process-based relationship between SIF and GPP such that information is passed directly via linear scaling parameters (i.e. the slope and intercept) to GPP and its related parameters, an approach that is yet to be evaluated against measurements or current theory.

The collective change in parameters results in an overall increase of LUE$_{\mathrm{GPP}}$ (Fig. B2) while APAR sees smaller, regionally

dependent changes (Fig. B4). We note that the reported LUE$_{\mathrm{GPP}}$ and APAR are annual mean values, as is the case with estimated model parameters. Seasonal variability also occurs and may need consideration when comparing to field-based studies. The posterior values for LUE$_{\mathrm{GPP}}$ are well within the expected physiological range with the theoretical maximum being 0.08 mol C mol photons$^{-1}$ (Waring et al., 2016). Overall, $V_{cmax}$ is the driving parameter behind changes in LUE$_{\mathrm{GPP}}$. We highlight that the latitudinal distribution of $V_{cmax}$ shows an increase in temperate and boreal zones following the SIF

assimilation (Fig. S19). The resulting zonal distribution of $V_{cmax}$ seems to be in closer agreement with independent studies using trait scaling, environmental scaling, and remote sensing retrieval methods that all show moderate values in the tropics and high values in the temperate zone (Ali et al., 2015; Alton, 2018; Walker et al., 2017). This remarkable result highlights the strength of a process-based SIF data assimilation system in estimating key biophysical parameters. This provides a pathway toward fully utilizing the information in SIF measurements. This also presents an opportunity to further evaluate our SIF-

optimized global patterns of LUE$_{\mathrm{GPP}}$, APAR and specific biophysical parameters against independent estimates.

Here, an advance is made on previous studies such that SIF is simulated in a process-based way rather than assuming simple linear scaling between SIF and GPP. This allows for the estimation of biophysical quantities (e.g. $C_{ab}$, $V_{cmax}$) based on the current theory of their relationship to SIF, with consideration of non-linear effects such as leaf and canopy radiative transfer and the relationship of quantum yields to photosynthetic rate. Our results also demonstrate that with the consideration of the

underlying processes, the model can increase SIF in some regions to better fit the observed data, but also show a decline in GPP (Figs. S17 and S18). This cannot occur when applying a linear scaling approach. Such patterns (e.g. wet tropical forests) are partly due to canopy composition and structure parameters such as leaf angle distribution, which can have a large effect on SIF but small effect on GPP, highlighting the point that SIF and GPP do not relate to biophysical parameters in the same way. Field-based studies applying a process-based data assimilation system with SIF, supplemented by other ecophysiological

measurements, should enable us to test the dynamical limitations on how SIF and GPP relate mechanistically.





Deficiencies in the model formulation and/or missing processes still limit the performance of the assimilation system. One example, investigated here, is the lack of time-varying biophysical parameters. Introducing time-varying parameters would improve the fit to observed SIF and in particular the large SIF values (Figs. 6 and C3). Introducing an empirical or mechanistic relationship between $C_{ab}$ and $V_{cmax}$ via their known relationship to nitrogen content (Evans, 1989) would also improve the

constraint SIF provides on GPP and better represent ecosystem function. Moreover, SCOPE is a 1D radiative transfer model and therefore may not effectively represent canopies with complex horizontal structure (e.g. open forest). More complex 3D models are under development (Gastellu-Etchegorry et al., 2017) however the high computational requirements may limit their application at the global scale. We note that further work is needed at both leaf and canopy scales to develop the model. The leaf level empirical formulation for NPQ also needs further testing as it partly determines how information is translated between

SIF and GPP via parameters like $V_{cmax}$. Finally, further work is needed to determine a mechanistic basis for drought stress effects on canopy SIF, such as that of Bayat et al. (2019), which can be implemented in BETHY-SCOPE.

There are other limitations to the present data assimilation system. Firstly, it's somewhat limited by use of prescribed LAI. This is exemplified by the regional assessment over the tropical forest of mainland south-east Asia (Fig. S10). We point out that the derived MODIS LAI and OCO-2 SIF show different seasonal patterns and that both are uncertain. Nevertheless, with

prescribed LAI the model is limited in its flexibility and cannot alter the shape of the seasonal cycle through the assimilation, resulting in a large posterior mismatch. This may also limit the ability of the model to simulate large SIF values. Secondly, the assimilation algorithm used cannot guarantee the global minimum of $J$ and hence optimal set of parameters, a problem for any local, gradient-based optimization. Thirdly, a number of potential sources of error are not accounted for in the error propagation. This means our uncertainty estimate for global GPP is likely to be an underestimate as it only accounts for uncertainties from

the parameters considered in Table A1. Inclusion of uncertainties in climate forcing and prescribed LAI would increase the uncertainty in global GPP although SIF would mediate this to some extent (Norton et al., 2018). Finally, systematic errors due to the instrument and retrieval errors, spatial sampling biases, and undersampling of diffuse light conditions as thick cloud prevents SIF retrieval may also need addressing in the future (Sun et al., 2018). Norton et al. (2018) did note, however, that one of the most important instrumental uncertainties arising from the correction of constant error artifacts in the SIF retrieval, did

not greatly contaminate results. Furthermore, spatial sampling limitations associated with OCO-2 may be overcome with the recently launched TROPOMI instrument that provides daily coverage of the complete Earth.

Future work should assess how SIF and other observational data may complement each other in constraining regions of model space. This would require explicit comparisons of observational constraints (e.g. atmospheric $CO_2$, carbonyl sulfide, or vegetation indices; EVI, FAPAR) using the same model. These data may be incorporated in a joint assimilation with SIF (e.g.

Peylin et al., 2016; Scholze et al., 2016) or used as independent data for validation purposes. Evaluating the SIF-optimized GPP patterns and resulting net terrestrial carbon flux will be particular focus of future work. Indeed the relatively high global GPP presented here would have implications for carbon–climate feedbacks, particularly for quantifying and modeling $CO_2$-fertilization and climate effects on the land carbon sink.





## 5   Conclusions

In this study we have presented the first application of satellite SIF to optimize parameters of a terrestrial biosphere model with a process-based model for SIF. We show, by comparing the model with satellite SIF observations within and outside of the calibration period, that there is substantial improvement in the predictive capability of the model following the optimization

with SIF. Despite this, there are still limitations of BETHY-SCOPE to match the high SIF values. This may be partly due to uncertainties in the prescribed LAI and a lack of temporal variability in biophysical parameters such as $C_{ab}$ and $V_{cmax}$. The SIF-optimized GPP is generally higher than the FLUXCOM GPP and TRENDY model average over the central tropics and temperate north. Following the assimilation there is a better match in the spatial contrast of the extra-tropical regions and the tropics compared with FLUXCOM GPP, and a better correlation with FLUXCOM GPP over North America and Europe that

have more training data. The use of SIF alters GPP by increasing $\text{LUE}_{\text{GPP}}$ across almost all ecosystems and altering APAR in regionally distinct ways. This study provides a significantly useful tool with which to improve our understanding of the global patterns of GPP. This may be extended by applying the model at flux tower sites, using additional satellite SIF data (e.g. GOSAT, GOME-2, TROPOMI), and assimilating other carbon cycle observations.

*Code and data availability.* The BETHY-SCOPE model code is available upon request from the authors. The OCO-2 satellite SIF data is

freely available at (insert doi here). Maps were produced using the freely available Panoply software (https://www.giss.nasa.gov/tools/panoply/).



## Appendix A: Spatially Dominant PFT in BETHY-SCOPE Model

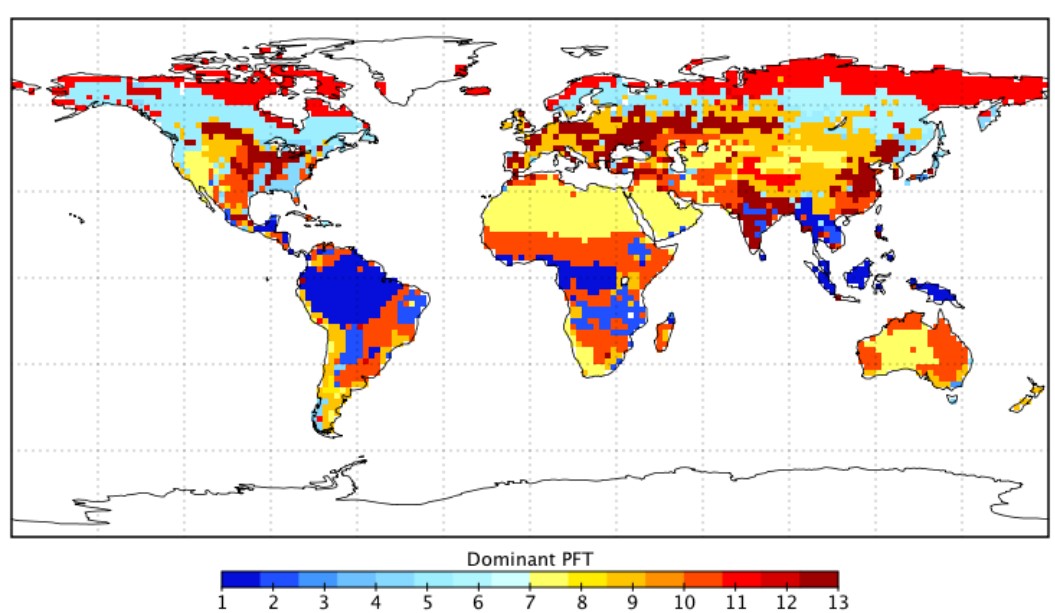

**Figure A1.** Spatially dominant PFT for each BETHY-SCOPE model grid cell.





## Appendix B: Posterior GPP Patterns

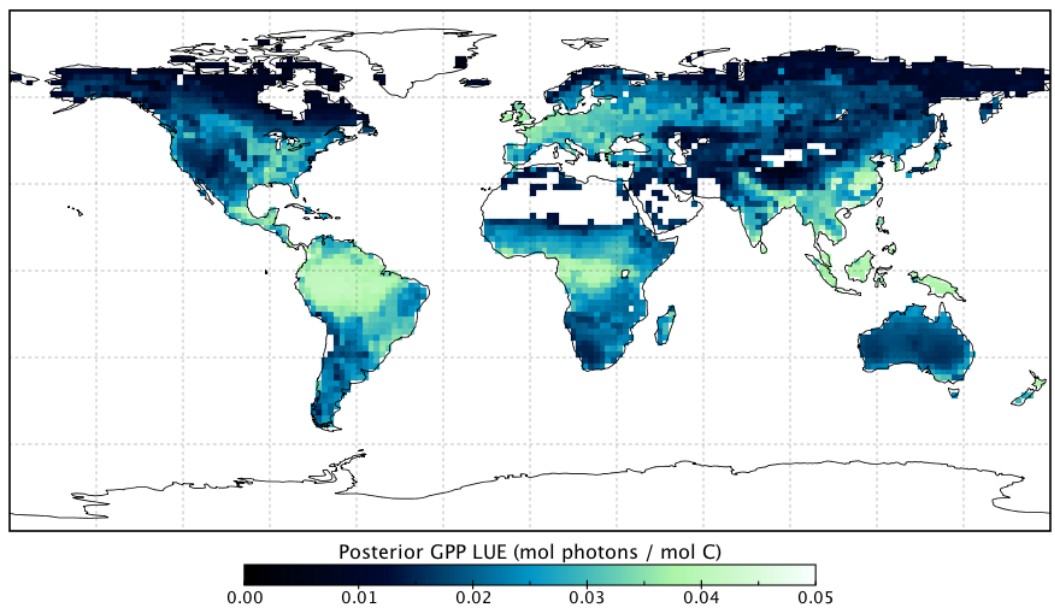

**Figure B1.** Posterior annual mean $\text{LUE}_{\text{GPP}}$ following the SIF assimilation.





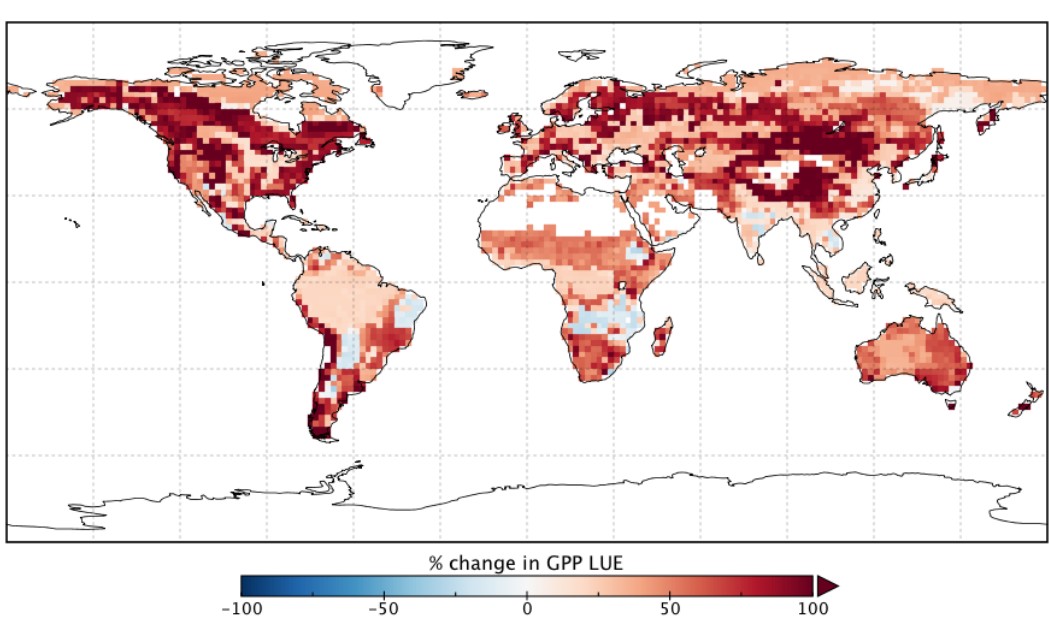

**Figure B2.** Percentage change in annual mean $LUE_{GPP}$ following the SIF assimilation.





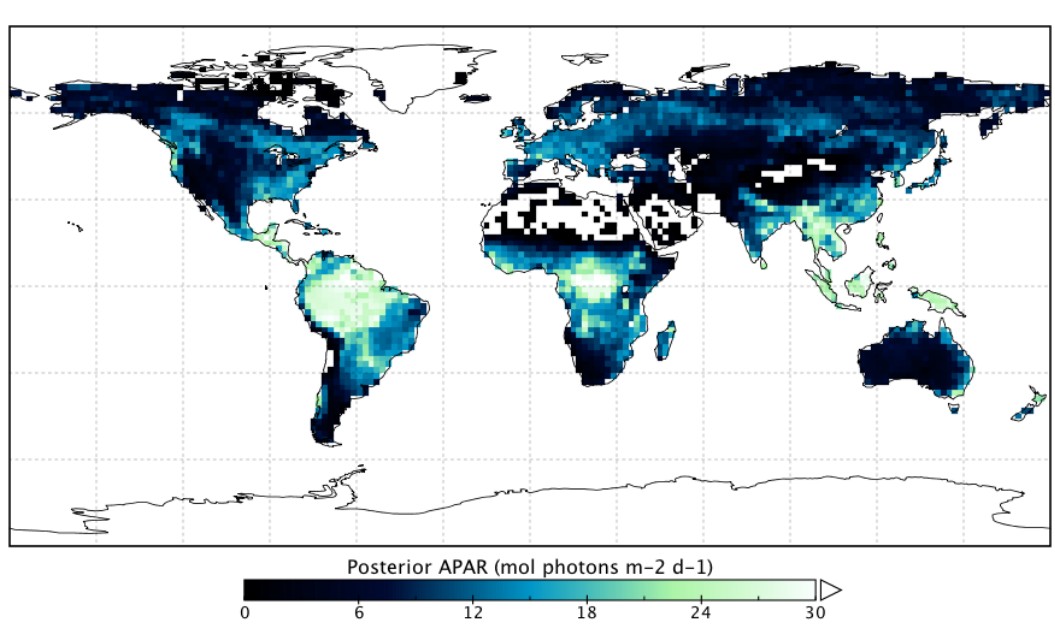

**Figure B3.** Posterior annual mean APAR following the SIF assimilation.





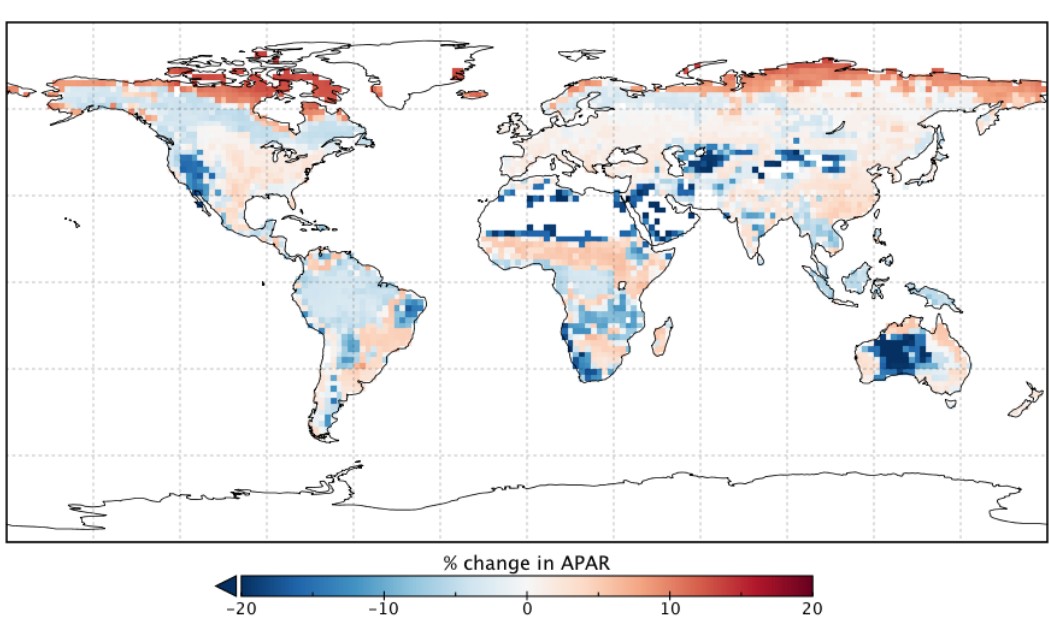

**Figure B4.** Percentage change in annual mean APAR following the SIF assimilation.





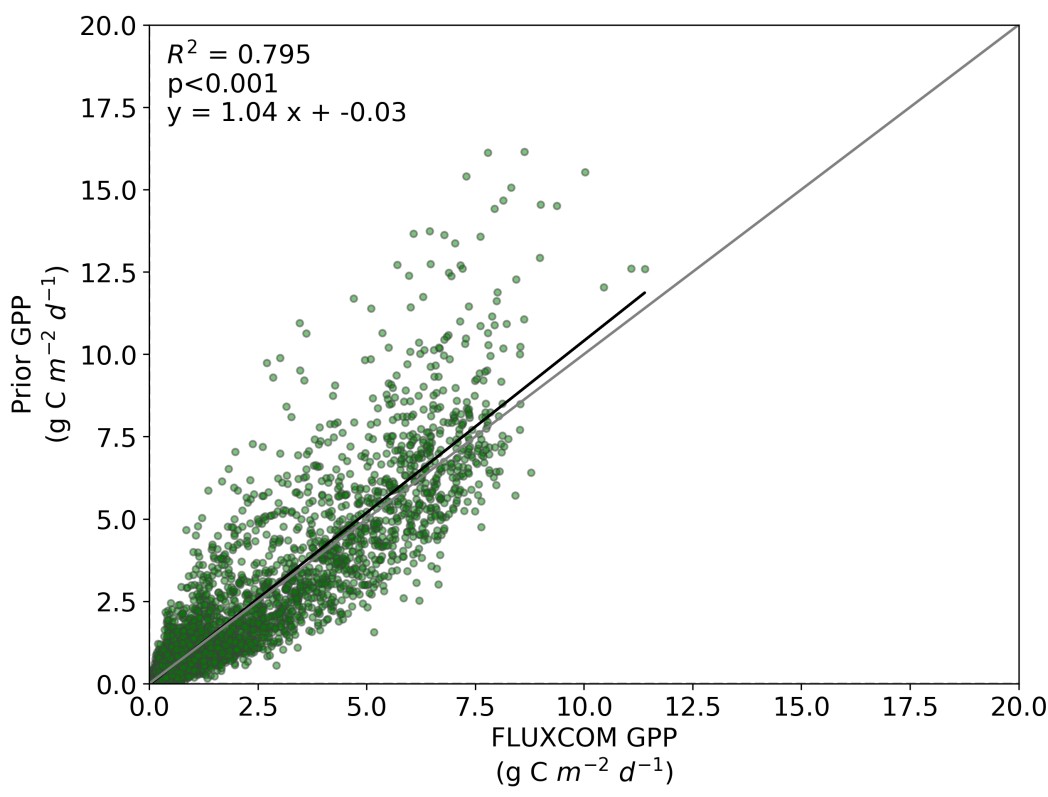

**Figure B5.** Comparison between FLUXCOM GPP and BETHY-SCOPE prior GPP over continental North America (between 24°N and 56°N).





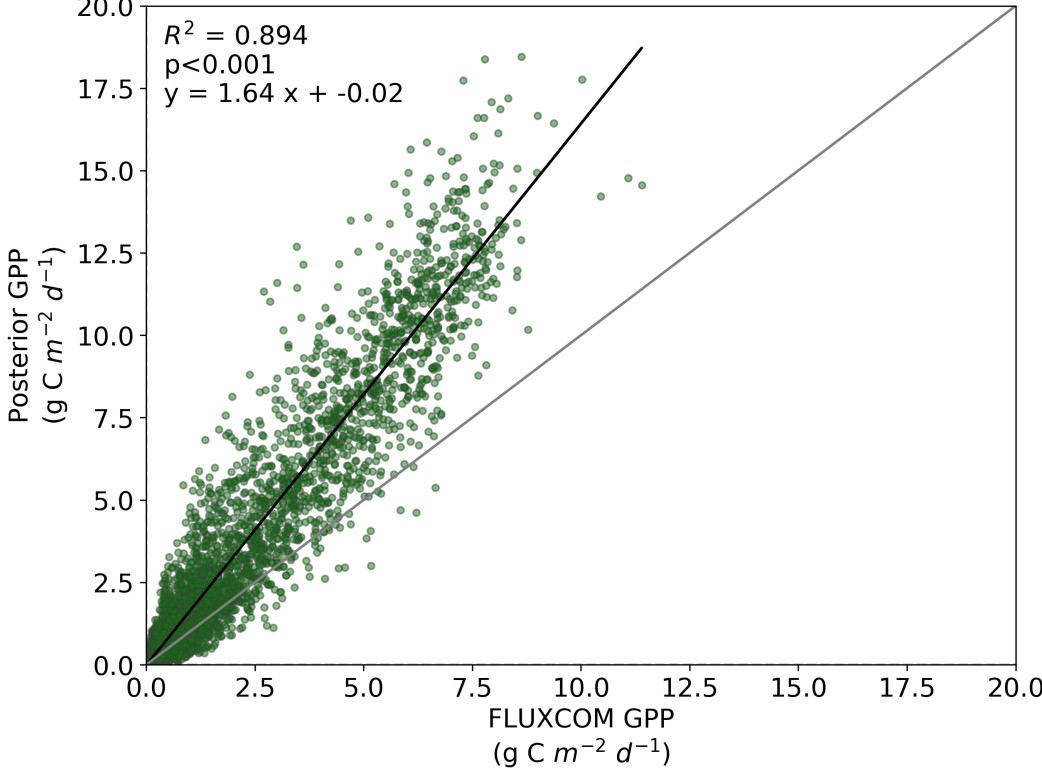

**Figure B6.** Comparison between FLUXCOM GPP and BETHY-SCOPE posterior GPP over continental North America (between 24°N and 56°N).




**Table A1.** BETHY-SCOPE process parameters along with their prior and optimized uncertainties following SIF constraint, represented as one standard deviation. Relative uncertainty reduction (i.e. effective constraint) is reported for the error propagation with low-resolution and high-resolution SIF observations. Units are: $V_{cmax}$, $\mu$mol($CO_2$) m$^{-2}$ s$^{-1}$; $a_{V_o,V_c}$, dimensionless ratio; $K_C$ and $K_O$, bar; $C_{ab}$, $\mu$g cm$^{-2}$; $C_{dm}$, g cm$^{-2}$; $C_{sm}$, dimensionless fraction; hc, m; leaf width, m. Footnotes: * The relative change is the parameter change in multiples of prior uncertainty. [a] Prior values based on Verhoef and Bach (2007). [b] Applies to PFTs TrEv, TrDec, TmpEv, TmpDec, EvShr, DecShr. [c] Applies to PFTs EvCn, DecCn, Tund. [d] Applies to PFTs C3Gr, C4Gr, Wetl, Crop. [e] Applies to PFTs TrEv, TrDec, TmpEv, TmpDec, EvCn, DecCn. [f] Applies to PFTs. EvShr, DecShr. [g] Applies to PFTs C3Gr, C4Gr, Tund, Wetl, Crop.

| Class | # | Description | Parameter | Prior | Posterior | Relative Change* | Effective Constraint (%) |
|---|---|---|---|---|---|---|---|
| LEAF PHYSIOLOGY | 1 | | $V_{cmax}$ (TrEv) | $60.0 \pm 12.0$ | $68.9 \pm 5.4$ | +0.74 | 55.1% |
| | 2 | | $V_{cmax}$ (TrDec) | $90.0 \pm 18.0$ | $40.7 \pm 8.6$ | -2.7 | 52.1% |
| | 3 | | $V_{cmax}$ (TmpEv) | $41.0 \pm 8.2$ | $28.7 \pm 6.6$ | -1.5 | 19.4% |
| | 4 | | $V_{cmax}$ (TmpDec) | $35.0 \pm 7.0$ | $81.6 \pm 4.8$ | +6.7 | 31.2% |
| | 5 | | $V_{cmax}$ (EvCn) | $29.0 \pm 5.8$ | $54.1 \pm 4.0$ | +4.3 | 30.0% |
| | 6 | Maximum | $V_{cmax}$ (DecCn) | $53.0 \pm 10.6$ | $73.5 \pm 7.5$ | +1.9 | 29.5% |
| | 7 | carboxylation rate | $V_{cmax}$ (EvShr) | $52.0 \pm 10.4$ | $51.8 \pm 8.0$ | -0.02 | 23.0% |
| | 8 | at 25°C | $V_{cmax}$ (DecShr) | $160.0 \pm 32.0$ | $89.1 \pm 26.3$ | -2.2 | 17.9% |
| | 9 | | $V_{cmax}$ (C3Gr) | $42.0 \pm 8.4$ | $101.6 \pm 6.8$ | +7.1 | 18.9% |
| | 10 | | $V_{cmax}$ (C4Gr) | $8.0 \pm 1.6$ | $15.8 \pm 1.3$ | +4.8 | 19.5% |
| | 11 | | $V_{cmax}$ (Tund) | $20.0 \pm 4.0$ | $30.4 \pm 3.6$ | +2.6 | 9.1% |
| | 12 | | $V_{cmax}$ (Wetl) | $20.0 \pm 4.0$ | $24.3 \pm 3.9$ | +1.1 | 1.6% |
| | 13 | | $V_{cmax}$ (Crop) | $117.0 \pm 23.4$ | $130.1 \pm 16.9$ | +0.6 | 28.0% |
| | 14 | Ratio of $V_{omax}$ to $V_{cmax}$ | $a_{V_o,V_c}$ | $0.220 \pm 0.0022$ | $0.219 \pm 0.0022$ | -0.4 | <1% |
| | 15 | Michaelis-Menten constant of Rubisco for $CO_2$ | $K_C$ | 350e-6 $\pm$ 17.5e-6 | 283e-6 $\pm$ 16.9e-6 | -3.8 | 3.6% |
| | 16 | Michaelis-Menten constant of Rubisco for $O_2$ | $K_O$ | $0.45 \pm 0.0225$ | $0.51 \pm 0.0224$ | +2.7 | <1% |





| | | | | | | |
|---|---|---|---|---|---|---|
| **LEAF COMPOSITION** | 17 | | $C_{ab}$ (TrEv) | $40.0 \pm 4.0$ | $28.5 \pm 3.5$ | -2.9 | 13.1% |
| | 18 | | $C_{ab}$ (TrDec) | $40.0 \pm 4.0$ | $17.8 \pm 3.6$ | -5.6 | 10.9% |
| | 19 | | $C_{ab}$ (TmpEv) | $30.0 \pm 3.0$ | $25.7 \pm 2.7$ | -1.4 | 9.7% |
| | 20 | | $C_{ab}$ (TmpDec) | $30.0 \pm 3.0$ | $37.7 \pm 2.4$ | +2.6 | 20.3% |
| | 21 | | $C_{ab}$ (EvCn) | $30.0 \pm 3.0$ | $18.7 \pm 2.5$ | -3.8 | 16.3% |
| | 22 | | $C_{ab}$ (DecCn) | $30.0 \pm 3.0$ | $29.1 \pm 2.7$ | -0.3 | 10.4% |
| | 23 | Chlorophyll $ab$ content | $C_{ab}$ (EvShr) | $20.0 \pm 2.0$ | $8.2 \pm 1.5$ | -5.9 | 25.7% |
| | 24 | | $C_{ab}$ (DecShr) | $20.0 \pm 2.0$ | $18.1 \pm 1.9$ | -0.9 | 7.5% |
| | 25 | | $C_{ab}$ (C3Gr) | $20.0 \pm 2.0$ | $18.4 \pm 1.8$ | -0.8 | 10.4% |
| | 26 | | $C_{ab}$ (C4Gr) | $20.0 \pm 2.0$ | $20.4 \pm 1.8$ | +0.2 | 11.5% |
| | 27 | | $C_{ab}$ (Tund) | $10.0 \pm 1.0$ | $12.3 \pm 0.7$ | +2.3 | 33.9% |
| | 28 | | $C_{ab}$ (Wetl) | $10.0 \pm 1.0$ | $11.1 \pm 0.9$ | +1.1 | 4.3% |
| | 29 | | $C_{ab}$ (Crop) | $40.0 \pm 4.0$ | $33.7 \pm 3.8$ | -1.6 | 4.5% |
| | 30 | Dry matter content | $C_{dm}$ | $0.0120 \pm 0.0020$ | $0.0207 \pm 0.0016$ | +4.4 | 18.9% |
| | 31 | Senescent material content | $C_{sm}$ | $0.00 \pm 0.01$ | $0.01 \pm 0.01$ | +1.2 | 1.3% |
| **CANOPY STRUCTURE** | 32 | | $LIDFa^{b}$ | $0.00 \pm 0.10$ | $-0.42 \pm 0.05$ | -4.2 | 54.0% |
| | 33 | Leaf inclination | $LIDFa^{c}$ | $-0.35 \pm 0.10$ | $-0.64 \pm 0.03$ | -2.9 | 71.8% |
| | 34 | distribution | $LIDFa^{d}$ | $-1.0 \pm 0.10$ | $-0.82 \pm 0.01$ | +1.8 | 90.0% |
| | 35 | function | $LIDFb^{b}$ | $-1.0 \pm 0.10$ | $-0.15 \pm 0.10$ | +8.5 | <2.4% |
| | 36 | parameters[a] | $LIDFb^{c}$ | $-0.15 \pm 0.10$ | $0.48 \pm 0.08$ | +6.3 | 20.0% |
| | 37 | | $LIDFb^{d}$ | $0.00 \pm 0.10$ | $1.00 \pm 0.06$ | +10.0 | 37.1% |
| | 38 | | hc[e] | $20.0 \pm 3.0$ | $20.1 \pm 3.0$ | 0.04 | <1% |
| | 39 | Vegetation height | hc[f] | $2.00 \pm 0.40$ | $1.97 \pm 0.40$ | -0.08 | <1% |
| | 40 | | hc[g] | $0.50 \pm 0.10$ | $0.05 \pm 0.10$ | -4.5 | <1% |
| | 41 | Leaf mesophyll structure | N | $0.10 \pm 0.01$ | $0.11 \pm 0.01$ | +1.2 | <1% |





**Table B1.** Estimated GPP per biome and latitudinal region, given as GPP rate and total annual GPP (in brackets). Biomes are defined by the spatially dominant PFT as shown in Fig. A1. The tropics are defined as the region between 30°S-30°N and the extratropics is as all latitudes outside of the tropics. The boreal region is defined as north of 54°N. The temperate region is defined as south of 30°S and 30-54°N.

| Biome or Region | Prior GPP, kgCm$^{-2}$yr$^{-1}$ (PgCyr$^{-1}$) | Posterior GPP, kgCm$^{-2}$yr$^{-1}$ (PgCyr$^{-1}$) | FLUXCOM GPP, kgCm$^{-2}$yr$^{-1}$ (PgCyr$^{-1}$) |
|---|---|---|---|
| TrEv | 3.5 (48.2) | 4.1 (56.0) | 2.2 (30.3) |
| TrDec | 2.0 (12.3) | 1.5 (9.5) | 1.2 (7.5) |
| TmpEv | 1.1 (0.4) | 1.1 (0.4) | 1.1 (0.4) |
| TmpDec | 0.9 (3.3) | 1.6 (6.1) | 1.0 (3.8) |
| EvCn | 0.6 (6.5) | 1.1 (11.1) | 0.7 (7.1) |
| DecCn | 0.7 (2.3) | 1.1 (3.5) | 0.5 (1.7) |
| EvShr | 0.1 (2.1) | 0.1 (2.6) | 0.3 (3.8) |
| DecShr | 0.5 (0.1) | 0.5 (0.1) | 0.4 (0.1) |
| C3Gr | 0.6 (11.4) | 1.2 (20.7) | 0.7 (12.2) |
| C4Gr | 0.7 (20.3) | 1.1 (31.3) | 0.8 (22.2) |
| Tund | 0.2 (1.8) | 0.2 (2.5) | 0.3 (2.7) |
| Wetl | 0.4 (0.3) | 0.5 (0.4) | 0.5 (0.5) |
| Crop | 1.4 (18.5) | 1.7 (22.5) | 0.8 (10.9) |
| Tropics | 1.62 (90.8 ±6.3) | 1.97 (110.0 ±3.4) | 1.21 (67.5) |
| Extratropics | 0.56 (36.8 ±2.4) | 0.86 (56.7 ±1.7) | 0.54 (35.6) |
| Boreal | 0.43 (10.1 ±0.7) | 0.68 (16.1 ±0.5) | 0.44 (10.1) |
| Temperate | 0.67 (26.6 ±1.8) | 1.02 (40.6 ±1.4) | 0.64 (25.8) |
| Tropics:Extratropics Ratio | 2.89 (2.47) | 2.29 (1.94) | 2.24 (1.90) |
| Tropics:Boreal Ratio | 3.77 (8.99) | 2.90 (6.83) | 2.75 (6.69) |
| Tropics:Temperate Ratio | 2.42 (3.41) | 1.93 (2.70) | 1.89 (2.66) |
| Temperate:Boreal Ratio | 1.56 (2.63) | 1.50 (2.52) | 1.45 (2.51) |

*Competing interests.* The authors declare that they have no conflicts of interest.





*Acknowledgements.* A. J. Norton was partly supported by an Australian Postgraduate Award from the Australian Government and a CSIRO OCE Scholarship. This research benefited from support provided by the ARC Centre of Excellence for Climate System Science (CE110001028). P. J. Rayner was supported by an Australian Research Council Fellowship (DP1096309). This research was undertaken with the assistance of resources and services from the National Computational Infrastructure (NCI), which is supported by the Australian Government. We

5   acknowledge the efforts of the TRENDY modelling group and thank them for supplying the TRENDY model data. We acknowledge the efforts of the OCO-2 science team and Christian Frankenberg for his assistance with the satellite SIF data.





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
