# Peer review of "Estimating global gross primary productivity using chlorophyll fluorescence and a data assimilation system with the BETHY-SCOPE model"

_Biogeosciences, 2019_

## Referee Comment (RC1) · Anonymous Referee #1 · 11 Apr 2019

Norton et al. "Estimating global gross primary productivity using chlorophyll fluorescence and a data assimilation system with the BETHY-SCOPE model".

GENERAL

This paper is a revised version of previous submission in 2018 with the same title. As a reviewer for both papers, I found the authors made helpful improvements but with new problems. In the earlier comments, I raised two main questions: First, the GPP is not effectively improved with SIF assimilation. Second, the GPP-SIF relations are not well

explained. Compared to the earlier version, this revision improves the second aspect but still fails to show a reasonable improvement in GPP.

The authors include more details about how GPP and SIF are connected in the model. In general, these two variables have some offsetting phases, because they share the same radiation energy. Such relationship explains why the posterior parameters reduce the high biases in SIF (Fig. 4) and consequently promote GPP (Fig. 10). However, compared to the 2018 paper, SIF is higher in the 2019 paper and is closer to observations (Fig. 4). Then why the GPP is much higher in this paper (167 Pg C yr-1) compared to earlier version (137 Pg C yr-1), instead of lower value? It shows that the SIF-GPP assimilation system may be arbitrary or casual about parameter adjustment.

The simulated GPP is much higher than present-day estimates from other studies/models. Results in Fig. 11 show that the 'improved' GPP is way too higher than the values from FLUXCOM and TRENDY. The authors claimed that GPP from FLUX-COM and TRENDY may be biased in tropics due to the limits in observations (Page 22). However, for mid-high latitudes (35-60ËŽN) in Northern Hemisphere where most of FLUXNET sites locate, the SIF-derived GPP values almost double the FLUXCOM. As a result, I think the assimilation system may have systematic biases, either from parameters (e.g., Fm', ÏŢp) or physical processes (e.g., the Equations 1-3), that degrade the values of this framework. In a word, the improvement of SIF does not effectively improve GPP.

---

## Referee Comment (RC2) · Anonymous Referee #2 · 21 May 2019

GPP is the largest carbon flux and constraining it is very important for understanding the terrestrial carbon sources and sinks. This paper presents a method to estimate the GPP in a data assimilation system based on the OCO2 SIF products. Compared to previous studies mainly based on the linear relationship between SIF and GPP, this paper adopts the process-based manner in which terrestrial biosphere model explicitly simulates the GPP and SIF. It is a new pathway to constrain GPP using the satellite SIF products. Also, there are some concerns about the results. I list them as follows: Several major concerns: (1) Actually I also noticed your previous online version

about this paper (Norton et al., 2018, Biogeosciences Discuss). I find that GPP can be largely increased by 31% in this new manuscript, while it was only increased by 7% in the previous one. So what's the difference in the background assimilation process? I carefully compare the prior parameter values in the Table A1, and find the only differences in the chlorophyll ab content (Cab). Is this the only difference? You mentioned that the Cab is set more in line with physiological understanding here (P6 Line 33-34). So what's the reference? If only tuning the prior Cab values makes the large difference, how to explain? The Cab value is only related to SIF not to GPP. (2) As you mentioned that the calculation of observation uncertainties is an important aspect of the data assimilation study. You calculate the observation uncertainty with a scale of 1/2 (Equation 4, P8 Line 11-20). How do you determine this scale? Sensitivity experiments? (3) P19 Line 9-10. You say that the changes in the posterior GPP can differ in sign and magnitude from the changes in posterior SIF. You explain it as a result of the non-linear effect in the process-based approach. First, you use the same SIF module as your previous manuscript (Norton et al., 2018, Biogeosciences Discuss)? It seems that the non-linear effect is not obvious in the previous one, but significant in this one, why? Second, If the nonlinear effect is obvious, how can we determine the GPP can be truly optimized? (4) P15 Line 1-3. You say that the SIFprior does not show this systematic underestimation, but has a poorer global fit (Fig. S3). If we look at the Figure S3, we can find that modeled SIF in lots of grids keep near constant (below 0.5). Or actually the scattering turns out the linear relationship is not statistical significant. But you show the $p < 0.001$, I think it is because you do not calculate the effective degrees of freedom, instead that you use the number of the points to calculate the p values. So the SIF module itself has large model errors. (5) In Section3.2, you show the results about the posterior parameters. You say "we can be more confident in parameters that see large reductions in uncertainty. Conversely, parameters with little reduction in uncertainty following optimization should be accepted cautiously." Actually, in the data assimilation, the uncertainty should be more or less reduced owing to the mathematics. But the reduction of the uncertainty does not mean the optimized

parameters are more accurate, because parameter optimization accounts for the LAI uncertainty, model structural uncertainty etc. In fact, posterior parameters can only partially improve the SIF simulation. In Figure 4, we can see the posterior SIF is more in line with the observed SIF. Therefore, the posterior parameters may be over-tuned. So is it possible to validate the posterior parameters based on the other datasets.

Specifics: (1) P1 Line 19. "(see Anav et al., 2015)/P2 Line 7 (see Baccour et al., 2015; ...)" ->remove the 'see" (2) P 6 Line 11 "Overall, the modelled link between SIF and GPP occurs via the above equations" ->actually it is not clear according to the above three equations. (3) P9 Line 11. Miss an "and". Should be "by the uncertainties in the observations Cd and model parameters Cx, respectively" (4) P11 Line 7-8. I can not clearly understand the sentence "... but forced by the respective monthly mean diurnal cycle such that a single diurnal cycle is simulated for each month" (5) Figure2 and Figure 3 can be presented in the same color bar. (6) P13 Line 14 "underestimate large observed SIF values > 0.5 W/m2/sr/um" (7) P15 Line 5 "This is largely because of observed SIF values that are slightly negative". Can it be shown in Figure 1 with the negative color bar. (8) Section 3.1.3 "A case with seasonally Varying Parameters" can be regarded as a discussion in the Section discussion. (9) P16 Line8-12 Is it possible add an equation here. (10) P 16 Line 17-18 "..with R2 increasing from 0.74 to 0.77 and the slope increasing from 0.67 to 0.71. This indicates that the systematic under-estimation of large observed SIF values may be improved." This conclusion is vague without the figures. (11) P20 the comparison between FLUXCOM and posterior GPP over the North America. The spatial correlation has an improvement with increasing correlation coefficient from 0.89 to 0.95. However, the amplitude is much larger than the FLUXCOM GPP. So it is improved or turns out poorer because you also mentioned the FLUXCOM GPP over north American and Europe may represent the actual GPP? (12) Maybe can adjust the orders of the Appendix figures. You first describe the Figs. B5 B6, then describe the Figs. B2 B3.. (13) P22 Line6. You say "In both of these studies an increase in tropical GPP was found", In Macbean et al., 2018, the posterior GPP seems a reduction?

---

## Author Comment (AC1) · 11 Jun 2019

We thank the referee for their comments. Here we provided a response to each comment. We hope that this new version is much improved. Note that original referee comments are enclosed in < > symbols.

While we have made the necessary changes following the referee comments, we have also rearranged and clarified parts of the discussion. We also noticed that we include one figure both in the manuscript and in the supplementary material. This has been

removed from the supplementary material.

<This paper is a revised version of previous submission in 2018 with the same title. As a reviewer for both papers, I found the authors made helpful improvements but with new problems. In the earlier comments, I raised two main questions: First, the GPP is not effectively improved with SIF assimilation. Second, the GPP-SIF relations are not well explained. Compared to the earlier version, this revision improves the second aspect but still fails to show a reasonable improvement in GPP. >

<The authors include more details about how GPP and SIF are connected in the model. In general, these two variables have some offsetting phases, because they share the same radiation energy. Such relationship explains why the posterior parameters reduce the high biases in SIF (Fig. 4) and consequently promote GPP (Fig. 10). However, compared to the 2018 paper, SIF is higher in the 2019 paper and is closer to observations (Fig. 4). Then why the GPP is much higher in this paper (167 Pg C yr-1) compared to earlier version (137 Pg C yr-1), instead of lower value? It shows that the SIF-GPP assimilation system may be arbitrary or casual about parameter adjustment. >

Given the concerns from RC1, it is perhaps helpful to point out the specific differences between our last assimilation setup and the present one. While it is not a focus of the paper to distinguish between these two, it is useful to present this in more detail here than is feasible in the manuscript. Here are the major differences in the assimilation setup between our current manuscript and the previous version to which the referee refers to: (i) The prior chlorophyll (Cab) parameters are set to be higher. The prior Cab parameters presented in the earlier version were too low (with a PFT average of 13 $\mu$g cm-2). This is not considered realistic as anything below $\sim$20 $\mu$g cm-2 suggests light interception is very low and will strongly limit photosynthesis (see Fig. 3.4 in Bjorkman, 1988); this is not expected under most natural conditions. This change in prior Cab values means our sensitivities (the slope of SIF with respect to Cab) were too large, as the expected change in SIF from Cab would be very large given it is strongly lightlimited. The effect of more realistic Cab values would be an increase in APAR and increase in GPP. We also set the 1-sigma prior uncertainty of all Cab parameters to be a consistent 10% of the mean value. (ii) The APAR provided to the photosynthesis module was total APAR in the previous version. In the SCOPE model, the total APAR represents the absorbed radiation by all canopy leaf elements. However, only green chlorophyll directs absorbed radiation to photochemistry. The model was therefore altered so that the green APAR was provided to the photosynthesis module, a change that has also been made to a more recent version of the SCOPE model. The effect of this more realistic model setup would be a decrease in APAR provided to GPP, hence a decrease in GPP. (iii) The version of Fluspect used in our last submission was not actually that of SCOPE v1.53. Therefore, we had to update the version of Fluspect in our model. Fluspect simulates the leaf level fluorescence and calculates the leaf level reflectance, transmittance and absorbances. The main issue was that the fluorescence quantum efficiency used in Fluspect (termed "fqe") was the same for photosystem I and II, but actually the values should be such that photosystem I is one fifth the value of of photosystem II. Therefore, the simulated SIF would have too high a contribution from PSI, which is not affected by physiological changes e.g. Vcmax.

These changes therefore include a change in model formulation and the prior parameters. We are confident that these changes make the model more realistic. The first two changes, i and ii, also have opposing effects on GPP, hence the prior GPP is similar between the current manuscript and previous version. The major difference is therefore in the sensitivities between SIF and the parameters, particularly Cab. This means our Jacobian matrix (H) is also different. The third change, iii, changes SIF but not GPP, but it will change the Jacobian matrix (H) as the contribution photosystem II to canopy SIF is larger and it is this photosystem that is regulated by physiological feedbacks. In layman's terms, the assimilation has different knobs and dials that it can use to minimise the cost function. If the model formulation or the Jacobian changes, these knobs and dials will change in size and strength, thus the posterior will also change.

There are a couple of notable differences in the posterior parameter set that probably cause the higher global GPP is in this version: (i) posterior Cab and leaf angle distribution (LIDF) parameters are different, with Cab being higher on average than our previous version (average of 13 PFTs is 22 $\mu$g cm-2 compared to 5 $\mu$g cm-2 in the last version), hence the new parameter set has higher APAR; (ii) the posterior Vcmax is slightly higher than the last version, with a PFT average of 61 $\mu$mol m-2 s-1 in the current version and 57 $\mu$mol m-2 s-1 in the previous version. This results in a higher APAR and higher LUE, hence higher GPP.

To address this, we have added a model version (now BETHY-SCOPE v1.1) to the methods to distinguish this version of BETHY-SCOPE from the previously submitted manuscript as well as the one used Norton et al. (2018) GMD paper (BETHY-SCOPE v1.0). We have also added a comment in the model description section of the Methods to highlight the changes: "In BETHY-SCOPE v1.1, the key changes are (i) the correction of an error in the Fluspect module where the fluorescence quantum efficiency (fqe) for PSI and PSII were set to be equal, while SCOPE v1.53 sets fqe for PSI to be one fifth that of PSII, and (ii) the leaf biochemistry module is now driven by green APAR (as mentioned above), rather than total APAR that is used in SCOPE v.153."

<The simulated GPP is much higher than present-day estimates from other studies/models. Results in Fig. 11 show that the 'improved' GPP is way too higher than the values from FLUXCOM and TRENDY. The authors claimed that GPP from FLUX-COM and TRENDY may be biased in tropics due to the limits in observations (Page 22). However, for mid-high latitudes (35-60E̊I̊L̊Z̊Ì̊Ñ̊N) in Northern Hemisphere where most of FLUXNET sites locate, the SIF-derived GPP values almost double the FLUX-COM. As a result, I think the assimilation system may have systematic biases, either from parameters (e.g., Fm',I̊L̊T İğp) or physical processes (e.g., the Equations 1-3), that degrade the values of this framework. In a word, the improvement of SIF does not effectively improve GPP. >

A recent review showed that global GPP is far from well-constrained as credible estimates range from 112-169 Pg C yr-1 (Anav et al., 2015). More recently, another SIF assimilation study by Macbean et al. (2018) produced a posterior global GPP of 166 Pg C yr-1, almost identical to our posterior estimate (although both have uncertainty ranges). Joiner et al. (2018) used GOME-2 satellite SIF and flux tower data to quantify global GPP and produced a global GPP of 140 Pg C yr-1 (in 2007). Other recent studies using different data suggest other estimates of 147 Pg C yr-1 (Badgley et al., 2018), 150-175 Pg C yr-1 (Welp et al., 2011), and 147 $\pm$ 19 Pg C yr-1 (Koffi et al., 2012). This large range of estimates reflects the lack of good direct observations and the inherent difficulty in quantifying global GPP. It also demonstrates that our posterior GPP estimate is not outside of other credible estimates, despite being at the higher end.

Overall, we do not state that our posterior global GPP wholly and completely "improved". In fact, what we state in our findings reflect our results quite directly e.g. in the abstract: "The SIF assimilation increases global GPP by 31% to 167 $\pm$ 5 Pg C yr-1 and shows an improvement in the global distribution of productivity relative to independent estimates, but a large difference in magnitude."

We respect the critique of the referee, but we do not find evidence that our GPP is beyond a credible range or that our parameters have systematic biases. Does the referee have some suggestion as to how they conclude our GPP is too high? Alternatively, is there are recommendation of what we should be changing?

We do not expect to produce a perfect or accurate global GPP estimate. In fact, we cannot even test whether our global GPP estimate is correct as there is no strict validation data for GPP at this scale. The FLUXCOM and TRENDY GPP products are used as a guide for the general patterns of GPP. As such, we assume the patterns of the FLUXCOM GPP are a useful check on our GPP patterns, but only in regions with plenty of flux sites, which is why we do the analysis in Figs. B5 and B6. We have made some changes to the discussion to make our interpretation clearer and to highlight the point that GPP estimates vary widely.

---

## Author Comment (AC2) · 11 Jun 2019

We thank the referee for their comments. Here we provided a response to each comment. We hope that this new version is much improved. Note that original referee comments are enclosed in < > symbols.

While we have made the necessary changes following the referee comments, we have also rearranged and clarified parts of the discussion. We also noticed that we include one figure both in the manuscript and in the supplementary material. This has been

removed from the supplementary material

<GPP is the largest carbon flux and constraining it is very important for understanding the terrestrial carbon sources and sinks. This paper presents a method to estimate the GPP in a data assimilation system based on the OCO2 SIF products. Compared to previous studies mainly based on the linear relationship between SIF and GPP, this paper adopts the process-based manner in which terrestrial biosphere model explicitly simulates the GPP and SIF. It is a new pathway to constrain GPP using the satellite SIF products.

Also, there are some concerns about the results. I list them as follows: Several major concerns:

(1) Actually I also noticed your previous online version about this paper (Norton et al., 2018, Biogeosciences Discuss). I find that GPP can be largely increased by 31% in this new manuscript, while it was only increased by 7% in the previous one. So what's the difference in the background assimilation process? I carefully compare the prior parameter values in the Table A1, and find the only differences in the chlorophyll ab content (Cab). Is this the only difference? >

Given that RC1 made a similar comment made, we provide the same response that we gave to RC1: Given the questions from both referees, it is perhaps helpful to point out the specific differences between our last assimilation setup and the present one. While it is not a focus of the paper to distinguish between these two, it is useful to present this in more detail here than is feasible in the manuscript. Here are the major differences in the assimilation setup between our current manuscript and the previous version to which the referee refers to: (i) The prior chlorophyll (Cab) parameters are set to be higher. The prior Cab parameters presented in the earlier version were too low (with a PFT average of 13 $\mu$g cm-2). This is not considered realistic as anything below $\sim$20 $\mu$g cm-2 suggests light interception is very low and will strongly limit photosynthesis (see Fig. 3.4 in Bjorkman, 1988); this is not expected under most natural conditions.

This change in prior Cab values means our sensitivities (the slope of SIF with respect to Cab) were too large, as the expected change in SIF from Cab would be very large given it is strongly light-limited. The effect of more realistic Cab values would be an increase in APAR and increase in GPP. We also set the 1-sigma prior uncertainty of all Cab parameters to be a consistent 10% of the mean value. (ii) The APAR provided to the photosynthesis module was total APAR in the previous version. In the SCOPE model, the total APAR represents the absorbed radiation by all canopy leaf elements. However, only green chlorophyll directs absorbed radiation to photochemistry. The model was therefore altered so that the green APAR was provided to the photosynthesis module, a change that has also been made to a more recent version of the SCOPE model. The effect of this more realistic model setup would be a decrease in APAR provided to GPP, hence a decrease in GPP. (iii) The version of Fluspect used in our last submission was not actually that of SCOPE v1.53. Therefore, we had to update the version of Fluspect in our model. Fluspect simulates the leaf level fluorescence and calculates the leaf level reflectance, transmittance and absorbances. The main issue was that the fluorescence quantum efficiency used in Fluspect (termed "fqe") was the same for photosystem I and II, but actually the values should be such that photosystem I is one fifth the value of of photosystem II. Therefore, the simulated SIF would have too high a contribution from PSI, which is not affected by physiological changes e.g. Vcmax.

These changes therefore include a change in model formulation and the prior parameters. We are confident that these changes make the model more realistic. The first two changes, i and ii, also have opposing effects on GPP, hence the prior GPP is similar between the current manuscript and previous version. The major difference is therefore in the sensitivities between SIF and the parameters, particularly Cab. This means our Jacobian matrix (H) is also different. The third change, iii, changes SIF but not GPP, but it will change the Jacobian matrix (H) as the contribution photosystem II to canopy SIF is larger and it is this photosystem that is regulated by physiological feedbacks. In layman's terms, the assimilation has different knobs and dials that it can use to minimise the cost function. If the model formulation or the Jacobian changes, these knobs

and dials will change in size and strength, thus the posterior will also change.

There are a couple of notable differences in the posterior parameter set that probably cause the higher global GPP is in this version: (i) posterior Cab and leaf angle distribution (LIDF) parameters are different, with Cab being higher on average than our previous version (average of 13 PFTs is 22 $\mu$g cm-2 compared to 5 $\mu$g cm-2 in the last version), hence the new parameter set has higher APAR; (ii) the posterior Vcmax is slightly higher than the last version, with a PFT average of 61 $\mu$mol m-2 s-1 in the current version and 57 $\mu$mol m-2 s-1 in the previous version. This results in a higher APAR and higher LUE, hence higher GPP.

To address this, we have added a model version (now BETHY-SCOPE v1.1) to the methods to distinguish this version of BETHY-SCOPE from the previously submitted manuscript as well as the one used Norton et al. (2018) GMD paper (BETHY-SCOPE v1.0). We have also added a comment in the model description section of the Methods to highlight the changes: "In BETHY-SCOPE v1.1, the key changes are (i) the correction of an error in the Fluspect module where the fluorescence quantum efficiency (fqe) for PSI and PSII were set to be equal, while SCOPE v1.53 sets fqe for PSI to be one fifth that of PSII, and (ii) the leaf biochemistry module is now driven by green APAR (as mentioned above), rather than total APAR that is used in SCOPE v.153."

<You mentioned that the Cab is set more in line with physiological understanding here (P6 Line 33-34). So what's the reference? > The reference is Bjorkman (1981). This study showed how chlorophyll concentrations below about 15-20 ug cm-2 cause steep declines in photosynthetic efficiency as only a very small fraction of light is intercepted at these concentrations. Optimal plant behaviour will act to prevent this.

<If only tuning the prior Cab values makes the large difference, how to explain? The Cab value is only related to SIF not to GPP.> Firstly, please refer to our extended response to your concern (1) above, as this addresses this comment as well. Briefly, we note that changing the prior Cab values causes a change in the Jacobian matrix

and will subsequently change how the assimilation minimises the cost function. We also note that Cab is actually strongly related to both SIF and GPP via APAR.

<(2) As you mentioned that the calculation of observation uncertainties is an important aspect of the data assimilation study. You calculate the observation uncertainty with a scale of 1/2 (Equation 4, P8 Line 11-20). How do you determine this scale? Sensitivity experiments? >

This scale, as we point out in the methods section 2.3.1, places our uncertainties roughly in the middle of the two extreme (and incorrect) ways of determining uncertainties. While rather arbitrary, in the end it's the reduced chi-squared statistic that determines whether we have selected appropriate uncertainties, including our choice of a $\frac{1}{2}$ scaling. We perhaps did not make this clear enough in section 2.3.1. So, we have added a specification in section 2.3.1 that the statistical tests used to test whether this is appropriate is the chi-squared test. This sentence now reads: "Statistical tests on the results, using the so-called reduced chi-squared statistic, allow us to test whether these observational uncertainties are consistent with other aspects of this data assimilation process, as outlined further below."

We also make a change in the results section where we report the posterior reduced chi-squared test. We make sure to refer back to the uncertainty calculation and the use of the $\frac{1}{2}$ scaling. This now reads: "The global $\chi_r^2$ fit is strongly reduced from 2.45 to 1.01. This is close to the optimal value of one, demonstrating the ability of the optimized model to fit the observed patterns of SIF and validating our chosen uncertainties as far as is practicable, including the choice of the scaling used to calculate observational uncertainties in Eq. \ref{eq:1}."

<(3) P19 Line 9-10. You say that the changes in the posterior GPP can differ in sign and magnitude from the changes in posterior SIF. You explain it as a result of the non-linear effect in the process-based approach. First, you use the same SIF module as your previous manuscript (Norton et al., 2018, Biogeosciences Discuss)? It seems that

the non-linear effect is not obvious in the previous one, but significant in this one, why?
>

In fact the same effect could occur in the previous manuscript and it did in some areas. We just hadn't highlighted it as a result. We believe it is an important result and a distinction of this process-based method that separates this study from those using empirical, linear relationships between SIF and GPP. We discuss this in detail in the Discussion section (P25, L26-35).

To clarify this, we have changed this line to: "We note that these changes in model GPP can differ in sign and magnitude from the changes in model SIF (see supplementary material Fig. S18 and S19). This can occur as SIF and GPP have differing sensitivities to the underlying parameters, a result of the process-based approach."

<Second, If the nonlinear effect is obvious, how can we determine the GPP can be truly optimized? > Just because the relationship is non-linear, this doesn't mean it is non-unique. Weather forecasting uses non-linear models for example. Many other inverse problems handle this. The problems arise if solutions are not unique. We do not see any suggestion that this is an issue. The posterior parameter values would likely reach into unphysical values if this was the case.

<(4) P15 Line 1-3. You say that the SIFprior does not show this systematic underestimation, but has a poorer global fit (Fig. S3). If we look at the Figure S3, we can find that modeled SIF in lots of grids keep near constant (below 0.5). Or actually the scattering turns out the linear relationship is not statistical significant. But you show the $p < 0.001$, I think it is because you do not calculate the effective degrees of freedom, instead that you use the number of the points to calculate the p values. So the SIF module itself has large model errors. > We calculate the reduced chi-squared statistic to evaluate the prior and posterior fit to the observations (Figs. S3 and S4), not the linear regression line or p-value. The reduced chi-squared statistic takes into account the number of degrees of freedom. The p-value you refer to is a simple p-value from

the linear regression. We used the linear regression simply to show what a linear regression fit looks like when fitted to the model vs observed data (i.e. is the slope near to one?), not as an actual test of the fit. We only report the reduced chi-squared statistic in the manuscript text. To ensure the p-value is not misleading other readers, we have removed it from the figure altogether.

<(5) In Section 3.2, you show the results about the posterior parameters. You say "we can be more confident in parameters that see large reductions in uncertainty. Conversely, parameters with little reduction in uncertainty following optimization should be accepted cautiously." Actually, in the data assimilation, the uncertainty should be more or less reduced owing to the mathematics. But the reduction of the uncertainty does not mean the optimized parameters are more accurate, because parameter optimization accounts for the LAI uncertainty, model structural uncertainty etc. In fact, posterior parameters can only partially improve the SIF simulation. In Figure 4, we can see the posterior SIF is more in line with the observed SIF. Therefore, the posterior parameters may be over-tuned. So is it possible to validate the posterior parameters based on the other datasets. > Yes, the referee is, in principle, correct. The uncertainty reductions are somewhat a measure of how exposed the parameters are to the observations (accounting for uncertainties of course). We cannot be sure that the parameters are accurate. This is relative though. Ultimately, a parameter that has no uncertainty reduction should not be accepted as being "constrained" by the data. A parameter that has a large uncertainty reduction should not be accepted as being accurate but can be relatively more accepted than one with no uncertainty reduction. As this statement is just relative, we have added "more" before cautiously.

Specifics:

<(1) P1 Line 19. "(see Anav et al., 2015)/P2 Line 7 (see Baccour et al., 2015; . . .)" ->remove the 'see'. > Done, thank you.

<(2) P 6 Line 11 "Overall, the modelled link between SIF and GPP occurs via the

above equations" ->actually it is not clear according to the above three equations. > If you consider that Ag is gross photosynthetic rate and Ðḏ′F is the fluorescence yield, then these equations outline how the model links the two. However, if the referee can indicate what is not clear about these equations, or how to make it better, we're open to making changes.

<(3) P9 Line 11. Miss an "and". Should be "by the uncertainties in the observations Cd and model parameters Cx, respectively". > Good catch thank you. Amended.

<(4) P11 Line 7-8. I cannot clearly understand the sentence "... but forced by the respective monthly mean diurnal cycle such that a single diurnal cycle is simulated for each month". > Yes, I can see how this might be confusing. This has been changed to "...a single, average diurnal cycle of meteorological forcing for each month is used to simulate photosynthesis and fluorescence. This allows the computation of SIF at the equivalent overpass time as the satellite data (1:00 - 2:00 p.m. local time)."

<(5) Figure2 and Figure 3 can be presented in the same color bar. > Does the referee suggest combining these two Figures into one and just using a single color bar? This can probably be done during the typesetting phase.

<(6) P13 Line 14 "underestimate large observed SIF values > 0.5 W/m2/sr/um". > The systematic underestimation of SIF only occurs beyond about 1.0 W m-2 sr-1 um-1. We can see from Fig. S4 that there a numerous spatiotemporal grid cells where model SIF is > 0.5 W m-2 sr-1 um-1. Hence, we have kept this as 1.0 W m-2 sr-1 um-1.

<(7) P15 Line 5 "This is largely because of observed SIF values that are slightly negative". Can it be shown in Figure 1 with the negative color bar. > Okay, we have changed Fig. 1 so that grid cells with negative values show up as white, and a new color bar that reflects this. Thank you for the suggestion.

<(8) Section 3.1.3 "A case with seasonally Varying Parameters" can be regarded as a discussion in the Section discussion. > Okay. We discussed this ourselves about

where best to place this. Note what comes next. There are new results presented based on this seasonally varying case in section 3.1.4 "Fit to the Seasonal Cycle", which would not make sense if the "case with seasonally Varying Parameters" was in the discussion. Therefore, we have kept these results where they are.

<(9) P16 Line8-12 Is it possible add an equation here. > Okay, we're happy to add the equation into the supplementary material, perhaps a better place as readers can find it if they're interested. We don't think it's pertinent to understanding our sensitivity test.

<(10) P 16 Line 17-18 "..with R2 increasing from 0.74 to 0.77 and the slope increasing from 0.67 to 0.71. This indicates that the systematic under- estimation of large observed SIF values may be improved." This conclusion is vague without the figures. > Ah yes, good point. We should have included them. We have added in the required figures to the supplementary material.

<(11) P20 the comparison between FLUXCOM and posterior GPP over the North America. The spatial correlation has an improvement with increasing correlation coefficient from 0.89 to 0.95. However, the amplitude is much larger than the FLUXCOM GPP. So it is improved or turns out poorer because you also mentioned the FLUXCOM GPP over north American and Europe may represent the actual GPP? > The simplest answer is that the spatiotemporal patterns are 'improved' but the magnitude gets 'worse' with respect to the FLUXCOM product. As discussed to the first referee, the true GPP is not known so we cannot be sure which estimate is correct. They're probably both wrong. So, we have removed the comment on "and thus where we expect it to better represent actual GPP". Nevertheless, we think this is the best comparison we can do between FLUXCOM and our SIF-based GPP considering the spatial scale. If the referee has a better idea of how to validate GPP we're open to suggestions.

<(12) Maybe can adjust the orders of the Appendix figures. You first describe the Figs. B5 B6, then describe the Figs. B2 B3. > Amended.

<(13) P22 Line6. You say "In both of these studies an increase in tropical GPP was

found", In Macbean et al., 2018, the posterior GPP seems a reduction? > Thanks for catching this. We are actually referring to the relative contribution to global GPP, in which their study sees the tropics increase relative to extratropics. Even so, in this paragraph our comment on our own results was not quite correct. We have actually changed this paragraph altogether, so this issue is no longer present.

References:

Anav et al. (2015), Spatiotemporal patterns of terrestrial gross primary production: A review. Rev. Geophys. 53, 785–818.

Bjorkman, O. (1981) Responses to different quantum flux densities, In: Physiological Plant Ecology I: Responses to physical environment, edited by Lange, O. et al., vol. 12A, pp. 57–107, Springer, Heidelberg, Berlin, and New York, https://doi.org/10.1111/aji.12612, 1981

Badgley, G., Anderegg, L. D. L., Berry, J. A., and Field, C. B.: An ecologically based approach to terrestrial primary production, EarthArXiv, https://doi.org/10.31223/osf.io/s6t3z, 2018.

Joiner et al. (2018) Estimation of Terrestrial Global Gross Primary Production (GPP) with Satellite Data-Driven Models and Eddy Covariance Flux Data, Remote Sensing, 10, doi:10.3390/rs10080000

Welp, L.R.; Keeling, R.F.; Meijer, H.A.J.; Bollenbacher, A.F.; Piper, S.C.; Yoshimura, K.; Francey, R.J.; Allison, C.E.; Wahlen, M. Interannual variability in the oxygen isotopes of atmospheric CO2 driven by El NinÌČo. Nature 2011, 477, 579–582.

Koffi et al. (2012) Atmospheric constraints on gross primary productivity and net ecosystem productivity: Results from a carbon-cycle data assimilation system, Global Biogeochemical Cycles, Vol. 26, GB1024, doi:10.1029/2010GB003900

Tramontana, G., Jung, M., Schwalm, C. R., Ichii, K., Camps-Valls, G., RaÌĄduly, B., Reichstein, M., Arain, M. A., Cescatti, A., Kiely, G., Merbold, L., Serrano-Ortiz, P., Sickert, S., Wolf, S., and Papale, D.: Predicting carbon dioxide and energy fluxes across global FLUXNET sites with regression algorithms, Biogeosciences, 13, 4291–4313, https://doi.org/10.5194/bg-13-4291-2016, 2016.

---

## Author Response (AR1)

RESPONSE TO REVIEWERS

Bg-2019-83:
Estimating global gross primary productivity using chlorophyll fluorescence and a data assimilation system with the BETHY-SCOPE model

General response to referees:

We thank the referees for their comments. Here we provide a response to each comment, including a general response to common referee comments. Following the referee comments we have made the necessary changes to the manuscript. We have also rearranged and clarified parts of the discussion. We also noticed that we include one figure both in the manuscript and in the supplementary material. This has been removed from the supplementary material. We are confident that this new version is improved.

Response to common referee comments. Note that referee comments are shown in black font while the author response is shown in blue font, and relevant changes are shown in red font.

RC1: compared to the 2018 paper, SIF is higher in the 2019 paper and is closer to observations (Fig. 4). Then why the GPP is much higher in this paper (167 Pg C yr-1) compared to earlier version (137 Pg C yr-1), instead of lower value?
RC2: I find that GPP can be largely increased by 31% in this new manuscript, while it was only increased by 7% in the previous one. So what's the difference in the background assimilation process?

Given the questions from both referees, it is perhaps helpful to point out the specific differences between our last assimilation setup and the present one. While it is not a focus of the paper to distinguish between these two, it is useful to present this in more detail here than is feasible in the manuscript. Here are the major differences in the assimilation setup between our current manuscript and the previous version to which the referee refers to:

(i)    The prior chlorophyll ($C_{ab}$) parameters are set to be higher. The prior $C_{ab}$ parameters presented in the earlier version were too low (with a PFT average of 13 µg cm$^{-2}$). This is not considered realistic as anything below ~20 µg cm$^{-2}$ suggests light interception is very low and will strongly limit photosynthesis (see Fig. 3.4 in Bjorkman, 1988); this is not expected under most natural conditions. This change in prior $C_{ab}$ values means our sensitivities (the slope of SIF with respect to $C_{ab}$) were too large, as the expected change in SIF from $C_{ab}$ would be very large given it is strongly light-limited. The effect of more realistic $C_{ab}$ values would be an increase in APAR and increase in GPP. We also set the 1-sigma prior uncertainty of all $C_{ab}$ parameters to be a consistent 10% of the mean value.

(ii)   The APAR provided to the photosynthesis module was total APAR in the previous version. In the SCOPE model, the total APAR represents the absorbed radiation by all canopy leaf elements. However, only green chlorophyll directs absorbed radiation to photochemistry. The model was therefore altered so that the green APAR was provided to the photosynthesis module, a change that has also been made to a more recent version of the SCOPE model. The effect of this more realistic model setup would be a decrease in APAR provided to GPP, hence a decrease in GPP.

(iii)  The version of Fluspect used in our last submission was not actually that of SCOPE v1.53. Therefore, we had to update the version of Fluspect in our model.

Fluspect simulates the leaf level fluorescence and calculates the leaf level reflectance, transmittance and absorbances. The main issue was that the fluorescence quantum efficiency used in Fluspect (termed "fqe") was the same for photosystem I and II, but actually the values should be such that photosystem I is one fifth the value of photosystem II. Therefore, the simulated SIF would have too high of a contribution from PSI, which is not affected by physiological changes e.g. $V_{cmax}$.

These changes therefore include a change in model formulation and the prior parameters. We are confident that these changes make the model more realistic. The first two changes, i and ii, also have opposing effects on GPP, hence the prior GPP is similar between the current manuscript and previous version. The major difference is therefore in the sensitivities between SIF and the parameters, particularly $C_{ab}$. This means our Jacobian matrix ($H$) is also different. The third change, iii, changes SIF but not GPP, but it will change the Jacobian matrix ($H$) as the contribution photosystem II to canopy SIF is larger and it is this photosystem that is regulated by physiological feedbacks. In layman's terms, the assimilation has different knobs and dials that it can use to minimise the cost function. If the model formulation or the Jacobian changes, these knobs and dials will change in size and strength, thus the posterior will also change.

There are a couple of notable differences in the posterior parameter set that probably cause the higher global GPP is in this version: (i) posterior $C_{ab}$ and leaf angle distribution (LIDF) parameters are different, with $C_{ab}$ being higher on average than our previous version (average of 13 PFTs is 22 µg cm$^{-2}$ compared to 5 µg cm$^{-2}$ in the last version), hence the new parameter set has higher APAR; (ii) the posterior $V_{cmax}$ is slightly higher than the last version, with a PFT average of 61 µmol m$^{-2}$ s$^{-1}$ in the current version and 57 µmol m$^{-2}$ s$^{-1}$ in the previous version. This results in a higher APAR and higher LUE, hence higher GPP.

To address this, we have added a model version (now BETHY-SCOPE v1.1) to the methods to distinguish this version of BETHY-SCOPE from the previously submitted manuscript as well as the one used Norton et al. (2018) GMD paper (BETHY-SCOPE v1.0). We have also added a comment in the model description section of the Methods to highlight the changes: "Between BETHY-SCOPE v1.0 used in \citet{Norton2018} and v1.1 used here, the key changes are (i) the correction of an error in the Fluspect module where the fluorescence quantum efficiency (fqe) for PSI and PSII were set to be equal, while SCOPE v1.53 sets fqe for PSI to be one fifth that of PSII, and (ii) the leaf biochemistry module is now driven by green APAR (as mentioned above), rather than total APAR that is used in SCOPE v.153." Also, under Methods section 2.2, we mention: "…the $C_{ab}$ parameters, which are assigned higher prior values than Norton et al. (2018), more in line with physiological understanding"

Finally, in the discussion section we include a detailed description of how this model, v1.1, differs from v1.10 used in Norton et al. (2018) GMD, described in the context of the differing uncertainty reductions reported. This outlines, in detail, the relative differences between the current study (with BETHY-SCOPE v1.1) and the GMD paper (with BETHY-SCOPE v1.0). We describe the effect of the new set of prior parameter values and the change in Fluspect model formulation.

Referee Comment 1:

GENERAL

This paper is a revised version of previous submission in 2018 with the same title. As a reviewer for both papers, I found the authors made helpful improvements but with new problems. In the earlier comments, I raised two main questions: First, the GPP is not effectively improved with SIF assimilation. Second, the GPP-SIF relations are not well explained. Compared to the earlier version, this revision improves the second aspect but still fails to show a reasonable improvement in GPP.

The authors include more details about how GPP and SIF are connected in the model. In general, these two variables have some offsetting phases, because they share the same radiation energy. Such relationship explains why the posterior parameters reduce the high biases in SIF (Fig. 4) and consequently promote GPP (Fig. 10). However, compared to the 2018 paper, SIF is higher in the 2019 paper and is closer to observations (Fig. 4). Then why the GPP is much higher in this paper (167 Pg C yr-1) compared to earlier version (137 Pg C yr-1), instead of lower value? It shows that the SIF-GPP assimilation system may be arbitrary or casual about parameter adjustment. Please refer to the author's general response section above.

The simulated GPP is much higher than present-day estimates from other studies/models. Results in Fig. 11 show that the 'improved' GPP is way too higher than the values from FLUXCOM and TRENDY. The authors claimed that GPP from FLUXCOM and TRENDY may be biased in tropics due to the limits in observations (Page 22). However, for mid-high latitudes (35-60ËŽN) in Northern Hemisphere where most of FLUXNET sites locate, the SIF-derived GPP values almost double the FLUXCOM. As a result, I think the assimilation system may have systematic biases, either from parameters (e.g., Fm',ÏT $_p$) or physical processes (e.g., the Equations 1-3), that degrade the values of this framework. In a word, the improvement of SIF does not effectively improve GPP.

A recent review showed that global GPP is far from well-constrained as credible estimates range from 112-169 Pg C yr$^{-1}$ (Anav et al., 2015). More recently, another SIF assimilation study by Macbean et al. (2018) produced a posterior global GPP of 166 Pg C yr$^{-1}$, almost identical to our posterior estimate (although both have uncertainty ranges). Joiner et al. (2018) used GOME-2 satellite SIF and flux tower data to quantify global GPP and produced a global GPP of 140 Pg C yr$^{-1}$ (in 2007). Other recent studies using different data suggest other estimates of 147 Pg C yr-1 (Badgley et al., 2018), 150-175 Pg C yr$^{-1}$ (Welp et al., 2011), and 147 ± 19 Pg C yr$^{-1}$ (Koffi et al., 2012). This large range of estimates reflects the lack of good direct observations and the inherent difficulty in quantifying global GPP. It also demonstrates that our posterior GPP estimate is not outside of other credible estimates, despite being at the higher end.

Overall, we do not state that our posterior global GPP wholly and completely "improved". In fact, what we state in our findings reflect our results quite directly e.g. in the abstract: "The SIF assimilation increases global GPP by 31% to 167 ± 5 Pg C yr$^{-1}$ and shows an improvement in the global distribution of productivity relative to independent estimates, but a large difference in magnitude."

We respect the critique of the referee, but we do not find evidence that our GPP is beyond a credible range or that our parameters have systematic biases. Does the referee have some suggestion as to how they conclude our GPP is too high? Alternatively, is there are recommendation of what we should be changing?

We do not expect to produce a perfect or accurate global GPP estimate. In fact, we cannot even test whether our global GPP estimate is correct as there is no strict validation data for GPP at this scale. The FLUXCOM and TRENDY GPP products are used as a guide for the general patterns of GPP. As such, we assume the patterns of the FLUXCOM GPP are a useful check on our GPP patterns, but only in regions with plenty of flux sites, which is why we do the analysis in Figs. B5 and B6. We have made some changes to the discussion to make our interpretation clearer and to highlight the point that GPP estimates vary widely.

To help address this comment, we have made changes to the Discussion section on p. 22. We clarify our interpretation of results and expand on the discussion of the high GPP number in the context of other studies.

Referee Comment 2:

GPP is the largest carbon flux and constraining it is very important for understanding the terrestrial carbon sources and sinks. This paper presents a method to estimate the GPP in a data assimilation system based on the OCO2 SIF products. Compared to previous studies mainly based on the linear relationship between SIF and GPP, this paper adopts the process-based manner in which terrestrial biosphere model explicitly simulates the GPP and SIF. It is a new pathway to constrain GPP using the satellite SIF products.

Also, there are some concerns about the results. I list them as follows: Several major concerns:

(1) Actually I also noticed your previous online version about this paper (Norton et al., 2018, Biogeosciences Discuss). I find that GPP can be largely increased by 31% in this new manuscript, while it was only increased by 7% in the previous one. So what's the difference in the background assimilation process? I carefully compare the prior parameter values in the Table A1, and find the only differences in the chlorophyll ab content (Cab). Is this the only difference? Please refer to our general response section above.

You mentioned that the Cab is set more in line with physiological understanding here (P6 Line 33-34). So what's the reference? The reference is Bjorkman (1981). This study showed how chlorophyll concentrations below about 15-20 ug cm$^{-2}$ cause steep declines in photosynthetic efficiency as only a very small fraction of light is intercepted at these concentrations. Optimal plant behaviour will act to prevent this, which is shown by Hirose and Werger (1987). We have modified the Discussion section to say: "into a range that is more likely to occur under typical conditions considering values <15 μg cm$^{-2}$ strongly reduce light interception and limit photosynthesis (Björkman, 1981; Hirose and Werger, 1987)".

If only tuning the prior Cab values makes the large difference, how to explain? The Cab value is only related to SIF not to GPP. Please refer to our general response section above. Briefly, we note that changing the prior $C_{ab}$ values causes a change in the Jacobian matrix and will subsequently change how the assimilation minimises the cost function. We also note that $C_{ab}$ is actually strongly related to both SIF and GPP via APAR.

(2) As you mentioned that the calculation of observation uncertainties is an important aspect of the data assimilation study. You calculate the observation uncertainty with a scale of 1/2 (Equation 4, P8 Line 11-20). How do you determine this scale? Sensitivity experiments? This scale, as we point out in the methods section 2.3.1, places our uncertainties roughly in the middle of the two extreme (and incorrect) ways of determining uncertainties. While rather arbitrary, in the end it's the reduced chi-squared statistic that determines whether we have selected appropriate uncertainties, including our choice of a ½ scaling.
We perhaps did not make this clear enough in section 2.3.1. So, we have added a specification in section 2.3.1 that the statistical tests used to test whether this is appropriate is the chi-squared test. This sentence now reads: "Statistical tests on the results, using the so-called reduced chi-squared statistic, allow us to test whether these observational uncertainties are consistent with other aspects of this data assimilation process, as outlined further below." We also make a change in the results section where we report the posterior reduced chi-squared test. We make sure to refer back to the uncertainty calculation and the use of the ½

 This now reads: "The global $\chi_r^2$ fit is strongly reduced from 2.45 to 1.01. This is close to the optimal value of one, demonstrating the ability of the optimized model to fit the observed patterns of SIF and validating our chosen uncertainties as far as is practicable, including the choice of the scaling used to calculate observational uncertainties in Eq. \ref{eq:1}."

(3) P19 Line 9-10. You say that the changes in the posterior GPP can differ in sign and magnitude from the changes in posterior SIF. You explain it as a result of the non-linear effect in the process-based approach. First, you use the same SIF module as your previous manuscript (Norton et al., 2018, Biogeosciences Discuss)? It seems that the non-linear effect is not obvious in the previous one, but significant in this one, why? In fact the same effect could occur in the previous manuscript and it did in some areas. We just hadn't highlighted it as a result. We believe it is an important result and a distinction of this process-based method that separates this study from those using empirical, linear relationships between SIF and GPP. We discuss this in detail in the Discussion section (P25, L26-35).
To clarify this, we have changed this line to: "We note that these changes in model GPP can differ in sign and magnitude from the changes in model SIF (see supplementary material Fig. S18 and S19). This can occur as SIF and GPP have differing sensitivities to the underlying parameters, a result of the process-based approach."

Second, If the nonlinear effect is obvious, how can we determine the GPP can be truly optimized?
Just because the relationship is non-linear, this doesn't mean it is non-unique. Weather forecasting uses non-linear models for example. Many other inverse problems handle this. The problems arise if solutions are not unique. We do not see any suggestion that this is an issue. The posterior parameter values would likely reach into unphysical values if this was the case.

(4) P15 Line 1-3. You say that the SIFprior does not show this systematic underestimation, but has a poorer global fit (Fig. S3). If we look at the Figure S3, we can find that modeled SIF in lots of grids keep near constant (below 0.5). Or actually the scattering turns out the linear relationship is not statistical significant. But you show the p < 0.001, I think it is because you do not calculate the effective degrees of freedom, instead that you use the number of the points to calculate the p values. So the SIF module itself has large model errors. We calculate the reduced chi-squared statistic to evaluate the prior and posterior fit to the observations (Figs. S3 and S4), not the linear regression line or p-value. The reduced chi-squared statistic takes into account the number of degrees of freedom. The p-value you refer to is a simple p-value from the linear regression. We used the linear regression simply to show what a linear regression fit looks like when fitted to the model vs observed data (i.e. is the slope near to one?), not as an actual test of the fit. We only report the reduced chi-squared statistic in the manuscript text. To ensure the p-value is not misleading other readers, we have removed it from the figure altogether.

(5) In Section 3.2, you show the results about the posterior parameters. You say "we can be more confident in parameters that see large reductions in uncertainty. Conversely, parameters with little reduction in uncertainty following optimization should be accepted cautiously." Actually, in the data assimilation, the uncertainty should be more or less reduced owing to the mathematics. But the reduction of the uncertainty does not mean the optimized parameters are more accurate, because parameter optimization accounts for the LAI uncertainty, model structural uncertainty etc. In fact, posterior parameters can only partially improve the SIF

simulation. In Figure 4, we can see the posterior SIF is more in line with the observed SIF. Therefore, the posterior parameters may be over-tuned. So is it possible to validate the posterior parameters based on the other datasets.

Yes, the referee is, in principle, correct. The uncertainty reductions are somewhat a measure of how exposed the parameters are to the observations (accounting for uncertainties of course). We cannot be sure that the parameters are accurate. This is relative though. Ultimately, a parameter that has no uncertainty reduction should not be accepted as being "constrained" by the data. A parameter that has a large uncertainty reduction should not be accepted as being accurate but can be relatively more accepted than one with no uncertainty reduction. As this statement is just relative, we have added "more" before cautiously.

Specifics:

(1) P1 Line 19. "(see Anav et al., 2015)/P2 Line 7 (see Baccour et al., 2015; …)" ->remove the 'see'. Done, thank you.

(2) P 6 Line 11 "Overall, the modelled link between SIF and GPP occurs via the above equations" ->actually it is not clear according to the above three equations. If you consider that $A_g$ is gross photosynthetic rate and $\Phi_F$ is the fluorescence yield, then these equations outline how the model links the two. However, if the referee can indicate what is not clear about these equations, or how to make it better, we're open to making changes.

(3) P9 Line 11. Miss an "and". Should be "by the uncertainties in the observations Cd and model parameters Cx, respectively". Good catch thank you. Amended.

(4) P11 Line 7-8. I cannot clearly understand the sentence "… but forced by the respective monthly mean diurnal cycle such that a single diurnal cycle is simulated for each month". Yes, I can see how this might be confusing. This has been changed to "…a single, average diurnal cycle of meteorological forcing for each month is used to simulate photosynthesis and fluorescence. This allows the computation of SIF at the equivalent overpass time as the satellite data (1:00 - 2:00 p.m. local time)."

(5) Figure2 and Figure 3 can be presented in the same color bar. Does the referee suggest combining these two Figures into one and just using a single color bar? This can probably be done during the typesetting phase.

(6) P13 Line 14 "underestimate large observed SIF values > 0.5 W/m2/sr/um". The systematic underestimation of SIF only occurs beyond about 1.0 W m-2 sr-1 um-1. We can see from Fig. S4 that there a numerous spatiotemporal grid cells where model SIF is > 0.5 W m-2 sr-1 um-1. Hence, we have kept this as 1.0 W m-2 sr-1 um-1.

(7) P15 Line 5 "This is largely because of observed SIF values that are slightly negative". Can it be shown in Figure 1 with the negative color bar. Okay, we have changed Fig. 1 so that grid cells with negative values show up as white, and a new color bar that reflects this. Thank you for the suggestion.

(8) Section 3.1.3 "A case with seasonally Varying Parameters" can be regarded as a discussion in the Section discussion.
Okay. We discussed this ourselves about where best to place this. Note what comes next.

There are new results presented based on this seasonally varying case in section 3.1.4 "Fit to the Seasonal Cycle", which would not make sense if the "case with seasonally Varying Parameters" was in the discussion. Therefore, we have kept these results where they are.

(9) P16 Line8-12 Is it possible add an equation here. Okay, we're happy to add the equation into the supplementary material, perhaps a better place as readers can find it if they're interested. We don't think it's pertinent to understanding our sensitivity test.

(10) P 16 Line 17-18 "..with R2 increasing from 0.74 to 0.77 and the slope increasing from 0.67 to 0.71. This indicates that the systematic under- estimation of large observed SIF values may be improved." This conclusion is vague without the figures. Ah yes, good point. We should have included them. We have added in the required figures to the supplementary material.

(11) P20 the comparison between FLUXCOM and posterior GPP over the North America. The spatial correlation has an improvement with increasing correlation coefficient from 0.89 to 0.95. However, the amplitude is much larger than the FLUXCOM GPP. So it is improved or turns out poorer because you also mentioned the FLUXCOM GPP over north American and Europe may represent the actual GPP? The simplest answer is that the spatiotemporal patterns are 'improved' but the magnitude gets 'worse' with respect to the FLUXCOM product. As discussed to the first referee, the true GPP is not known so we cannot be sure which estimate is correct. They're probably both wrong. So, we have removed the comment on "and thus where we expect it to better represent actual GPP". Nevertheless, we think this is the best comparison we can do between FLUXCOM and our SIF-based GPP considering the spatial scale. If the referee has a better idea of how to validate GPP we're open to suggestions.

(12) Maybe can adjust the orders of the Appendix figures. You first describe the Figs. B5 B6, then describe the Figs. B2 B3. Amended.

(13) P22 Line6. You say "In both of these studies an increase in tropical GPP was found", In Macbean et al., 2018, the posterior GPP seems a reduction? Thanks for catching this. We are actually referring to the relative contribution to global GPP, in which their study sees the tropics increase relative to extratropics. Even so, in this paragraph our comment on our own results was not quite correct. We have actually changed this paragraph altogether, so this issue is no longer present.

[revised manuscript text omitted]

**Model-Observed Mismatch**

[Figure]

**Figure S8.** Annual total mismatch between the observed SIF and prior model SIF ($SIF_{prior}$).

[Figure]

**Figure S9.** Annual total mismatch between the observed SIF and posterior model SIF ($SIF_{post}$).

[Figure]

**Figure S10.** Model versus observed seasonal amplitude of SIF for the prior model (left), SIF-optimized model (middle), and SIF-optimized model with seasonally varying $C_{ab}$ and $V_{cmax}$ parameters (right). Shown on each plot is a 1:1 line (grey) and linear regression line (blue) with the associated equation. Also shown is the mean ratio between the model and observed seasonal amplitude.

**Regional Model-Observed Differences**

[Figure]

**Figure S11.** Regional patterns of SIF (left), GPP (center) and LAI (right) over mainland south-east Asia only for model grid cells with TrEv as the dominant PFT (see Fig. S12). The shading represents one sigma spread of data points in the selected region and month.

[Figure]

**Figure S12.** Model grid points selected for the regional analysis of mainland south-east Asia tropical forest.

[Figure]

**Figure S13.** Regional patterns of SIF (left), GPP (center) and LAI (right) over North America croplands only for model grid cells with Crop as the dominant PFT (see Fig. S14). The shading represents one sigma spread of data points in the selected region and month.

[Figure]

**Figure S14.** Model grid points selected for the regional analysis of North American croplands.

[Figure]

**Figure S15.** Regional patterns of SIF (left), GPP (center) and LAI (right) over north Africa savanna only for model grid cells with C4 grass as the dominant PFT (see Fig. S16). The shading represents one sigma spread of data points in the selected region and month.

[Figure]

**Figure S16.** Model grid points selected for the regional analysis of north African savanna.

**Seasonal Adjustment of Parameters**

This equation determines the seasonal adjustment made to $V_{cmax}$ and $C_{ab}$ parameters for the sensitivity test in Section 3.1.3. The posterior parameter value is denoted by $x_{post}$ and the seasonally adjusted parameter value is denoted by $x_{adjusted}$. Note that for highly seasonal PFTs including deciduous trees and shrubs, C3 and C4 grasses, and crops, the amplitude factor (f) is set to 50% of the mean (f=0.5), while for all other PFTs the amplitude is set to 10% (f=0.1). The DOY is the day of the year, between 1 and 365.

$$x_{adjusted}(DOY) = x_{post} + f\ x_{post}\ sin(2\pi/365(DOY - 81)) \tag{1}$$

$$x_{adjusted}(DOY) = x_{post} + f\ x_{post}\ sin(2\pi/365(DOY - 264)) \tag{2}$$

**Spatiotemporal Patterns of GPP, SIF, $V_{cmax}$ and $C_{ab}$**

[Figure]

**Figure S17.** Percentage change in annual mean GPP rate for 2015 following optimization with SIF relative to GPP$_{prior}$.

[Figure]

**Figure S18.** Change in annual mean SIF for 2015 following optimization with SIF relative to SIF$_{prior}$.

[Figure]

**Figure S19.** Latitudinal average of mapped maximum carboxylation capacity at 25°C, $V_{cmax}$, parameter values for the prior and posterior cases.

[Figure]

**Figure S20.** Latitudinal average of mapped chlorophyll content, $C_{ab}$, parameter values for the prior and posterior cases.

---

## Author Response (AR2)

AUTHOR RESPONSE

Bg-2019-83:
Estimating global gross primary productivity using chlorophyll fluorescence and a data assimilation system with the BETHY-SCOPE model

Many thanks to the editor for their helpful technical comments. We have made the recommended changes. Note that we have also:
- Added in the missing details under "code and data availability" for the OCO-2 SIF data.
- Changed a sub-heading in the appendix changed $LUE_P$ to $LUE_{GPP}$. The acronym $LUE_P$ was used in an earlier version of the manuscript, but was superseded by $LUE_{GPP}$.

Many thanks.